# Cell competition removes segmental aneuploid cells from *Drosophila* imaginal disc-derived tissues based on ribosomal protein gene dose

Zhejun Ji[†], Jacky Chuen, Marianthi Kiparaki[‡], Nicholas Baker*

Department of Genetics, Albert Einstein College of Medicine, Bronx, United States

**Abstract** Aneuploidy causes birth defects and miscarriages, occurs in nearly all cancers and is a hallmark of aging. Individual aneuploid cells can be eliminated from developing tissues by unknown mechanisms. Cells with ribosomal protein (*Rp*) gene mutations are also eliminated, by cell competition with normal cells. Because *Rp* genes are spread across the genome, their copy number is a potential marker for aneuploidy. We found that elimination of imaginal disc cells with irradiation-induced genome damage often required cell competition genes. Segmentally aneuploid cells derived from targeted chromosome excisions were eliminated by the RpS12-Xrp1 cell competition pathway if they differed from neighboring cells in *Rp* gene dose, whereas cells with normal doses of the *Rp* and *eIF2γ* genes survived and differentiated adult tissues. Thus, cell competition, triggered by differences in *Rp* gene dose between cells, is a significant mechanism for the elimination of aneuploid somatic cells, likely to contribute to preventing cancer.

**\*For correspondence:**
nicholas.baker@einsteinmed.org

**Present address:** [†]State Key Laboratory of Stem Cell and Reproductive Biology, Institute of Zoology, Chinese Academy of Sciences, Beijing 100101, China; [‡] Biomedical Sciences Research Center 'Alexander Fleming', Vari, Greece

**Competing interests:** The authors declare that no competing interests exist.

## Introduction

Aneuploidy (gain or loss of whole chromosomes resulting in an abnormal karyotype) is a hallmark of spontaneous abortions and birth defects and observed in virtually every human tumor (*Hassold and Hunt, 2001*; *Hanahan and Weinberg, 2011*; *López-Otín et al., 2013*). It was suggested over 100 years ago that aneuploidy contributes to cancer development (*Boveri, 1914*). Aneuploidy can change the copy number of important oncogenes and tumor suppressors, cause stress due to gene expression imbalance, and promote further genetic instability (*Naylor and van Deursen, 2016*; *Rutledge and Cimini, 2016*; *Chunduri and Storchová, 2019*; *Ben-David and Amon, 2020*; *Zhu et al., 2018*). Mouse models of chromosome instability that result in aneuploidy are oncogenic (*Foijer et al., 2008*; *Baker et al., 2009*; *Mukherjee et al., 2014*). *Drosophila* cells with chromosome instability undergo a p53-independent death, but can form tumors if their apoptosis is prevented (*Dekanty et al., 2012*; *Gerlach and Herranz, 2020*; *Morais da Silva et al., 2013*).

Because aneuploidy is thought to be detrimental to normal cells, aneuploid cells arising sporadically in vivo should, as a rule, grow poorly (*Sheltzer and Amon, 2011*). Studies of yeast carrying extra chromosomes reveal a stress response in these cells, thought to result from the cumulative mismatch in levels of many proteins that interact in the cell, which inhibits growth (*Torres et al., 2007*; *Zhu et al., 2018*; *Terhorst et al., 2020*).

Increasing evidence points to the capacity of normal tissues to recognize and eliminate aneuploid cells (*Hook, 1981*; *van Echten-Arends et al., 2011*; *Bazrgar et al., 2013*; *Pfau et al., 2016*; *Santaguida et al., 2017*). Array Comparative Genome Hybridization detects mosaic aneuploidy in as many as 60% of normal human embryos, which can nonetheless develop into healthy babies without birth defects or evidence of aneuploid cells, suggesting their elimination (*Greco et al., 2015*). In mice, chimeric embryos can be constructed using both normal diploid cells and cells with a high rate

**eLife digest** Aneuploid cells emerge when cellular division goes awry and a cell ends up with the wrong number of chromosomes, the tiny genetic structures carrying the instructions that control life's processes. Aneuploidy can lead to fatal conditions during development, and to cancer in an adult organism.

A safety mechanism may exist that helps the body to detect and remove these cells. Yet, exactly this happens is still poorly understood: in particular, it is unclear how cells manage to 'count' their chromosomes.

One way they could do so is through the ribosomes, the molecular 'factories' that create the building blocks required for life. In a cell, every chromosome carries genes that code for the proteins (known as Rps) forming ribosomes. Aneuploidy will alter the number of Rp genes, and in turn the amount and type of Rps the cell produces, so that ribosomes and the genes for Rps could act as a 'readout' of aneuploidy. Ji et al set out to test this theory in fruit flies.

The first experiment used a genetic manipulation technique called site-specific recombination to remove parts of chromosomes from cells in the developing eye and wing. Cells which retained all their Rp genes survived, while those that were missing some usually died – but only when the surrounding cells were normal. In this situation, healthy cells eliminated their damaged neighbours through a process known as cell competition. A second experiment, using radiation as an alternative method of damaging chromosomes, also gave similar results.

The work by Ji et al. reveals how the body can detect and eliminate aneuploid cells, potentially before they can cause harm. If the same mechanism applies in humans, boosting cell competition may, one day, helps to combat diseases like cancer.

of aneuploidy due to treatment with reversine, an inhibitor of the spindle assembly checkpoint. The reversine-treated cells are actively eliminated from the chimeric embryos, which can develop into morphologically normal adult mice from which reversine-treated cells have been eliminated (*Bolton et al., 2016*). Other observations point to the loss of aneuploid cells in other biological processes. For example, the cortex of normal mouse embryos contains as many as 30% aneuploid cells, but only ~1% are detected by 4 months post-partum, suggesting selective loss of the aneuploid fraction (*Andriani et al., 2016*).

The mechanisms of recognition and removal of aneuploid cells are still poorly understood. In mouse tissues, cells with complex karyotypes may be recognized by the immune system (*Santaguida et al., 2017*). In *Drosophila*, clones of segmentally aneuploid cells (cells with loss or gain of chromosome segments) can survive development and differentiate in the adult abdomen, but their representation decreases as more genetic material is lost, whereas cells carrying extra genetic material are less affected (*Ripoll, 1980*). Clonal loss of heterozygosity is also tolerated in the abdomens of DNA repair pathway mutants, and studies with genetic markers indicate that this frequently represents loss of substantial chromosome segments (*Baker et al., 1978*).

The *Drosophila* adult abdomen derives from larval histoblasts. In the head and thorax, which develop instead from the larval imaginal discs, there is evidence that aneuploid cells undergo apoptosis. DNA damage following ionizing irradiation rapidly leads to apoptosis but is followed by a smaller amount of delayed apoptosis that is independent of p53 and Chk2 and therefore unlikely to reflect unrepaired DNA damage (*Brodsky et al., 2004*; *Wichmann et al., 2006*). A similar biphasic response is seen after mitotic breakage of dicentric chromosomes, and in this case, the delayed, p53-independent cell death only occurs in genotypes likely to lead to aneuploid products (*Titen and Golic, 2008*). Accordingly, it is suggested that post-irradiation apoptosis independent from p53 also represents removal of aneuploid cells that arise following DNA repair, and that 'cell competition' may provide the p53-independent mechanism (*McNamee and Brodsky, 2009*).

The term 'cell competition' was originally coined to describe the elimination of *Drosophila* cells heterozygous for mutant alleles of ribosomal protein genes (*Rp* genes)(*Morata and Ripoll, 1975*). Most Rp's are essential, even to the individual cell, so that homozygosity for $Rp^-$ mutations is rapidly lethal, whereas $Rp^{+/-}$ heterozygotes are viable and fertile, although slow growing with minor morphological defects such as thin adult bristles (*Marygold et al., 2007*). By contrast to their whole

animal viability, individual $Rp^{+/-}$ heterozygous cells or clones are actively eliminated from mosaic *Drosophila* tissues (*Morata and Ripoll, 1975*; *Simpson, 1979*). This involves apoptosis specific to $Rp^{+/-}$ heterozygous cells near to $Rp^{+/+}$ cells (*Morata and Ripoll, 1975*; *Simpson, 1979*; *Moreno et al., 2002*; *Li and Baker, 2007*). The defining feature of cell competition is therefore the elimination of cells based on their difference from other neighboring cells rather than based on their intrinsic properties (*Morata and Ripoll, 1975*; *Baker, 2020*).

The 80 eukaryotic Rp's are mostly encoded by single copy genes transcribed by RNA polymerase II, and are dispersed throughout the genome in both humans and in *Drosophila* (*Uechi et al., 2001*; *Marygold et al., 2007*). Accordingly, aneuploidy and other large-scale genetic changes will usually affect *Rp* gene dose. Since Rp proteins are required stoichiometrically for ribosome assembly, which generally stalls when any one Rp is limiting, imbalanced *Rp* gene dose can perturb ribosome biogenesis (*de la Cruz et al., 2015*). This provides an almost perfectly suited mechanism to serve as an indicator of unbalanced chromosome content (*McNamee and Brodsky, 2009*). Cell competition may thus have evolved to recognize and remove cells with large-scale genetic changes such as aneuploidy, recognized on the basis of their mis-matched Ribosomal protein (*Rp*) gene complements. In this view, cells heterozygous for point mutations in *Rp* genes are eliminated because they mimic larger genetic changes.

A stress response pathway that is activated by *Rp* mutations in *Drosophila* has recently been described, and is required for *Rp* point mutated cells to undergo cell competition (*Baillon et al., 2018*; *Kale et al., 2018*; *Lee et al., 2018*; *Ji et al., 2019*; *Blanco et al., 2020*). In $Rp^{+/-}$ genotypes, RpS12, an essential, eukaryote-specific component of the ribosomal Small Subunit, is required to activate expression of Xrp1, a rapidly evolving AT-hook, bZip domain transcription factor (*Lee et al., 2018*; *Ji et al., 2019*; *Blanco et al., 2020*). Although *rpS12* null mutations are homozygously lethal, a particular point mutation, *rpS12*$^{G97D}$, appears defective only for the cell competition aspect of RpS12 function. Homozygotes for the *rpS12*$^{G97D}$ mutation are viable, showing only minor effects on morphology and longevity, yet *rpS12*$^{G97D}$ prevents elimination of $Rp^{+/-}$ cells by cell competition (*Kale et al., 2018*; *Ji et al., 2019*). A key target of RpS12 appears to be the putative transcription factor Xrp1, because Xrp1 protein is barely detected in wild type cells but significantly elevated in $Rp^{+/-}$ wing discs. Xrp1 controls most of the phenotype of $Rp^{+/-}$ cells, including their reduced translation (*Lee et al., 2018*; *Ji et al., 2019*; *Blanco et al., 2020*). *Xrp1* mutants have negligible effect in wild-type backgrounds, and normal lifespan (*Baillon et al., 2018*; *Lee et al., 2018*; *Mallik et al., 2018*). In response to RpS12 and Xrp1 activities, $Rp^{+/-}$ cells both grow more slowly than surrounding $Rp^{+/+}$ cells and are also actively eliminated by apoptosis that occurs where $Rp^{+/-}$ cells and $Rp^{+/+}$ cells meet (*Baillon et al., 2018*; *Lee et al., 2018*; *Ji et al., 2019*). Why apoptosis occurs at these interfaces in particular is not certain. A role for innate immune pathway components has been proposed (*Meyer et al., 2014*), and different levels of oxidative stress response in $Rp^{+/-}$ cells and $Rp^{+/+}$ cells (*Kucinski et al., 2017*) or local induction of autophagy have also been suggested (*Nagata et al., 2019*).

Here, we test the hypothesis that cell competition specifically removes cells with aneuploidies that result in loss of *Rp* genes in *Drosophila* imaginal discs. We show that, as hypothesized previously (*McNamee and Brodsky, 2009*), most of the p53-independent cell death that follows irradiation resembles cell competition genetically. We then use a targeted recombination method to investigate the fate of somatic cells that acquire large-scale genetic changes directly, and confirm that it is cell competition that removes cells heterozygous for large deletions, when they include ribosomal protein genes. By contrast, when ribosomal protein genes are unaffected, or when the cell competition pathway is inactivated genetically, cells carrying large deletions remain largely uncompeted, proliferate, and contribute to adult structures. Thus, cell competition is a highly significant mechanism for elimination of aneuploid somatic cells. We discuss how cell competition to remove aneuploid cells could play a role preventing tumor development in humans.

## Results

### Potential competition of irradiated cells

The idea that cell competition removes aneuploid cells was suggested by studies of p53-independent cell death following chromosome breakage or ionizing irradiation (*Titen and Golic, 2008*;

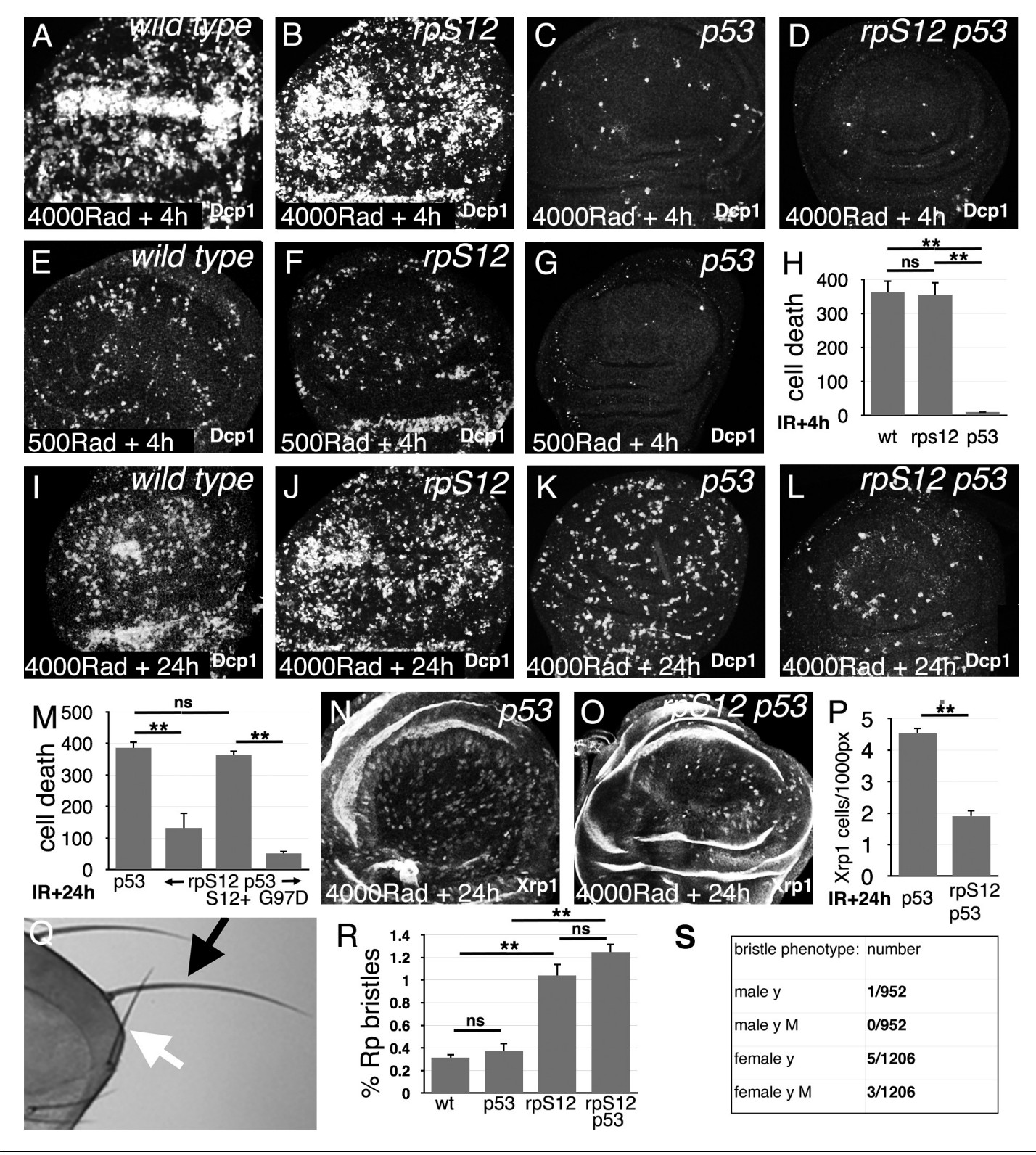

**Figure 1.** Role of cell competition after irradiation. Panels A-G and I-L show third instar wing imaginal discs labeled to detect the activated Dcp1 caspase in apoptotic cells at the indicated times post-irradiation. (**A**) Extensive cell death follows within 4 hr after gamma-irradiation (4000 Rads). (**B**) Little change is seen in the *rpS12*^G97D mutant. (**C**) Most of the acute cell death is p53-dependent, consistent with a DNA damage response. (**D**) A small amount of cell death persists in the *rpS12*^G97D *p53*⁻ double mutant. (**E**) Cell death is reduced to more quantifiable levels 4 hr after a lower radiation

*Figure 1 continued on next page*

Figure 1 continued

dose (500 Rad). (F) This DNA damage-induced cell death is not significantly affected by the $rpS12^{G97D}$ mutation. (G) Most cell death 4 hr after irradiation is p53-dependent. (H) Quantification of cell death (numbers of cells per wing pouch) 4 hr following irradiation with 500 Rad. ns – difference not statistically significant (p>0.05). ** - difference highly significant (p<0.01). N = 6 for each genotype. (I) Cell death 24 hr after irradiation (4000 Rad). (J) Comparable levels of cell death in $rpS12^{G97D}$ mutants. (K) Although cell death appears reduced in $p53$ mutant wing discs 24 hr after irradiation compared to the wild type control (4000 Rad), this p53-independent cell death is substantially increased compared to 4 hr after irradiation (compare panel C). (L) p53-independent cell death is reduced in the $rpS12^{G97D}$ $p53^-$ double mutant compared to the $p53^-$ mutant. (M) Cell death quantification in $p53$ and $rpS12^{G97D}$ $p53^-$ wing discs, and in wing discs from $rpS12^{G97D}$ $p53^-$ larvae carrying genomic transgenes encoding either wild type $rpS12$ or $rpS12^{G97D}$. The results show that between 66% and 86% of p53-independent cell death was RpS12-dependent. We did not quantify cell death in wild-type and $rpS12^{G97D}$ wing discs because of the large number and aggregation of dead cells. ns – difference not statistically significant (p>0.05). ** - difference highly significant (p<0.01). N: 10 ($p53^-$); 13 ($rpS12^{G97D}$ $p53^-$); 9 (P{$rpS12^+$} $rpS12^{G97D}$ $p53^-$); 8 (P{$rpS12^{G97D}$} $rpS12^{G97D}$ $p53^-$). (N) Twenty-four hr after irradiation, $p53$ mutant wing discs exhibit cells expressing nuclear Xrp1 in a pattern similar to that of dying cells. We were unable to double-label with anti-Xrp1 and anti-active Dcp1 simultaneously because both are rabbit antisera. n: 33 ($p53^-$); 26 ($rpS12^{G97D}$ $p53^-$). (O) Fewer Xrp1-expressing cells were seen in $rpS12^{G97D}$ $p53^-$ double mutant wing discs. (P) Quantification of Xrp1 expression. Most (58%) of the p53-independent Xrp1 expression 24 hr post-irradiation was RpS12-dependent. ** - difference highly significant (p<0.01). (Q) Irradiated adult flies with a short, thin scutellar bristle (white arrow, compare normal contralateral bristle – black arrow) like those typical of $Rp^{+/-}$ mutant flies. (R) The frequency of sporadic $Rp$-like bristles increases 3.33x and 4x on the thoraces of $rpS12^{G97D}$ and $rpS12^{G97D}$ $p53^-$ flies (respectively) where cell competition is inhibited. ns – difference not statistically significant (p>0.05). ** - difference highly significant (p<0.01). N = 3 sets of 100 flies for each genotype. (S) Frequencies of $y$ bristles found on the thoraces of ~2000 $y/+$; $rpS12^{G97D}$ female and $+/Y$; $rpS12^{G97D}$ male flies following irradiation (1000 rad) in the mid-third larval instar. The preponderance of $y$ bristles in females suggests that induced $y$ mutations typically affect other genes including essential genes. The occurrence of phenotypically $y$ $M$ bristles in females is consistent with deletions extending at least from the $y$ locus to the nearest $Rp$ locus, $RpL36$. Statistics: Significance was assessed using t-test (panel P) or one-way ANOVA with the Holm procedure for multiple comparisons (panels H,M,R). ** = p<0.01. ns = not significant. Genotypes: A, (E, I) $w^{11-18}$ B, (F, J) $rpS12^{G97D}$ (C,G,K,N) $p53^{5A-1-4}$ D, (L, O) $rpS12^{G97D}$ $p53^{5A-1-4}$.

The online version of this article includes the following source data for figure 1:

**Source data 1.** Source Data for *Figure 1H,M,P,R*.

---

*McNamee and Brodsky, 2009*). Importantly, in *Drosophila* cell competition of $Rp^{+/-}$ cells does not depend on p53 (*Kale et al., 2015*). If the model was correct, it would be expected that the p53-independent apoptosis that follows irradiation would depend on the genes recently discovered to be required for cell competition.

We first confirmed the previous findings (*Wichmann et al., 2006*; *McNamee and Brodsky, 2009*). Irradiating third-instar larvae resulted in rapid induction of cell death in the wing imaginal disc that was largely p53-dependent (*Figure 1A,C,D,E,G,H*). The p53-dependent cell death, attributable to the DNA-damage response, was not much affected by the $rpS12^{G97D}$ mutation that interferes with cell competition (*Figure 1B,F,H*). While total cell death tailed off with time, p53-independent cell death increased around 18–24 hr post-irradiation, as reported previously (*Wichmann et al., 2006*; *McNamee and Brodsky, 2009*; *Figure 1I–L*). As expected for cell competition, 24 hr after irradiation, p53-independent cell death was reduced by 66% in the $rpS12^{G97D}$ $p53$ double mutant compared to the $p53$ mutant alone (*Figure 1K–M*). To exclude the possibility that other genetic background differences were responsible, $rpS12$ function was restored to the $rpS12^{G97D}$ $p53$ double mutant strain using a P element transgene encoding the wild-type $rpS12$ gene, and this restored p53-independent cell death (*Figure 1M*). Notably, a genomic transgene encoding the $rpS12^{G97D}$ cell competition-defective allele did not, leading to 86% less p53-independent cell death than the wild-type $rpS12$ transgene (*Figure 1M*). Thus, 66–86% of the p53-independent apoptosis was RpS12-dependent and might represent cell competition.

Nuclear Xrp1 protein, which is only at low levels in control wing imaginal discs, was detected in scattered cells throughout p53 mutant wing discs 24 hr after irradiation, similar to the distribution of dying cells (*Figure 1N*). Strikingly, most (58%) of this Xrp1 expression was $rpS12$-dependent, similar to the $rpS12$-dependency of p53-independent cell death itself (*Figure 1O–P*). This is also as expected for cell competition, which is mediated through the induction of Xrp1 expression (*Baillon et al., 2018*; *Lee et al., 2018*; *Ji et al., 2019*).

Radiation-damaged cells that have reduced $Rp$ gene dose would be expected to differentiate short, thin bristles in adults, as reported previously in studies of DNA repair mutants (*Baker et al., 1978*), in studies of ionizing radiation (*McNamee and Brodsky, 2009*), and when aneuploidy is induced by mutation of spindle assembly checkpoint genes (*Dekanty et al., 2012*). We found $Rp^{+/-}$-like thoracic bristles at a frequency of ~1/300 following irradiation of either wild type or p53 mutant

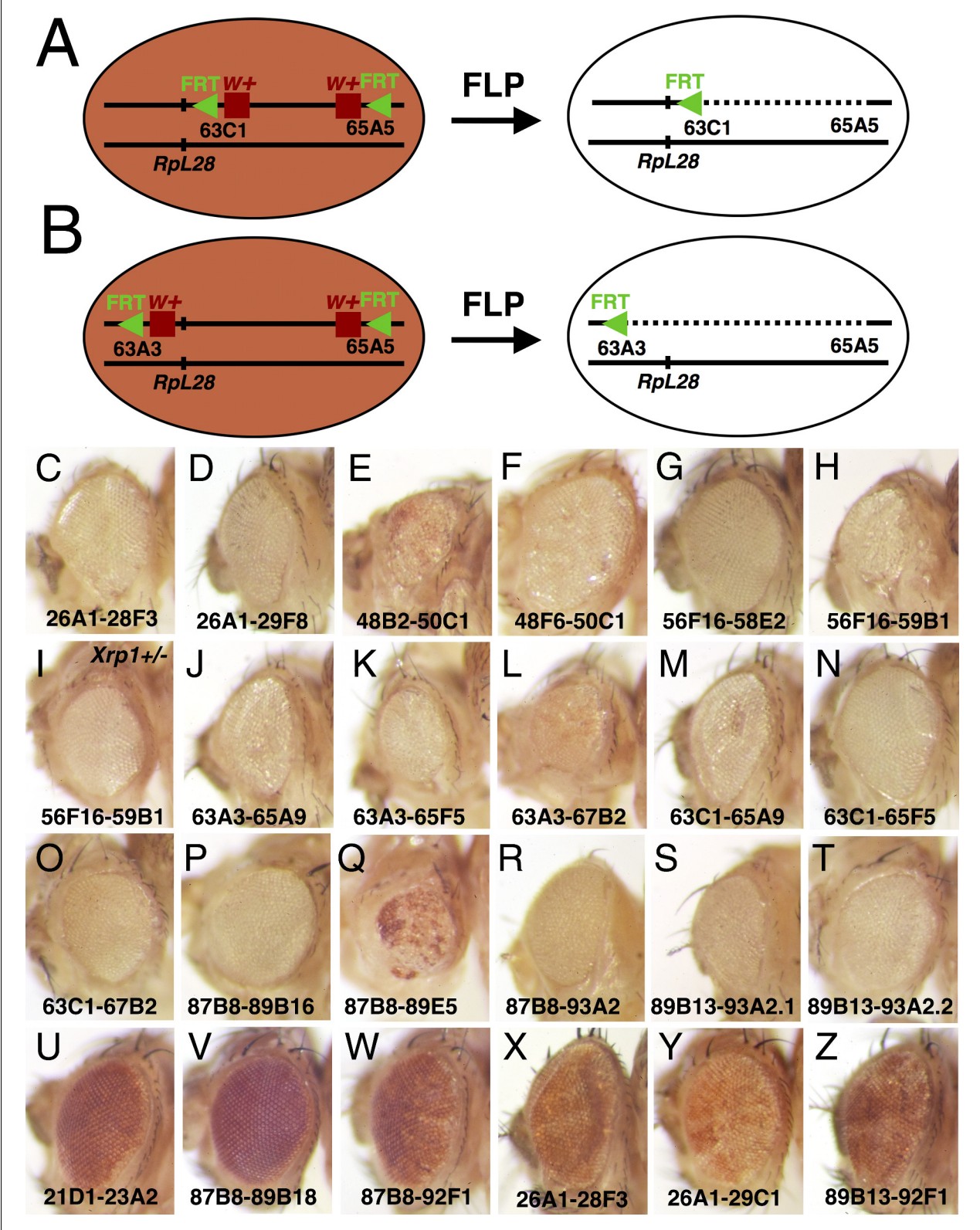

**Figure 2.** Generating segmental aneuploidy with Flp-*FRT* recombination. (**A**) Segmental aneuploidy can be generated in a mosaic fashion using FLP-*FRT*. At left is a cell carrying two transposable elements encoding *w⁺* and *FRT*, arranged in cis on one chromosome arm, in this case at chromosome bands 63C1 and 65A9, respectiely. (**B**) FLP-mediated recombination between *FRT* sites excising the intervening sequences. In this example, where the *w⁺* genes lie between the *FRT* sites, both are excised resulting in a loss of eye pigmentation. The *RpL28* gene at chromosome band 63B14 lies outside

*Figure 2 continued on next page*

*Figure 2 continued*

the deletion and is unaffected. (B) A comparable recombination between elements at chromosome bands 63A3-65A9 also deletes the *RpL28* locus, so that the resulting segmentally aneuploid cells are heterozygously deleted for this gene. These cartoons show recombination in G1-phase of the cell cycle. In the G2-phase configuration, recombination between non-homologous FRT sites on the chromatids can occur leading to a deleted chromatid and a chromatid bearing 3 *FRT* insertion elements and a duplication of the intervening region. Such genotypes are substrates for further FLP recombination to the parental or deleted state, but sometimes we see them persist in adults and an example is shown in **Figure 3D** (*Titen et al.,* **2020**). Panels C-Z show adult eyes in the presence of eyFlp, which drives recombination close to completion in the eye and head, with the chromosome positions of parental $w^+$ *FRT* insertions indicated. Panels C-T show genotypes where eyFlp recombined most eye cells, Panels U-Z illustrate genotypes where it did not. Strains that were poor substrates for Flp, either retained the parental eye color in the presence of eyFlp, or produce a salt and pepper pattern of very small clones that is indicative of excision occurring only late in development once large cell numbers are present. See **Supplementary file 1** for more details and further genotypic information. (C) *y w eyFlp; P{XP}$^{d08241}$ PBac{WH}$^{f04888}$/+;* (D) *y w eyFlp; P {XP}$^{d08241}$ PBac{WH}$^{f00857}$/+.* This recombination deletes the *RpL36A* and *RpS13* genes; (E) *y w eyFlp; P{XP}$^{d09761}$ PBac{WH}$^{f00157}$/+.* This recombination deletes the *RpS11* gene. Note the particularly small size of the recombinant heads; (F) *y w eyFlp; P{XP}$^{d09417}$ PBac{WH}$^{f00157}$/+.* (G) *y w eyFlp; P {XP}$^{d02302}$ PBac{WH}$^{f04349}$/+;* (H) *y w eyFlp; P{XP}$^{d02302}$ PBac{WH}$^{f00464}$/+.* This recombination deletes the *RpS16* and *RpS24* genes. Note the particularly small size of the recombinant heads; (I) *y w eyFlp; P{XP}$^{d02302}$ PBac{WH}$^{f00464}$/Xrp1$^{m2-73}$.* Eye size is partially rescued by heterozygosity for *Xrp1*. (J) *y w eyF; PBac{WH}$^{f01922}$ P{XP}$^{d02570}$ /+.* This recombination deletes the *RpL28* gene. (K) *y w eyF; PBac{WH}$^{f01922}$ P{XP}$^{d02813}$ /+.* This recombination deletes the *RpL28* and *RpL18* genes; (L) *y w eyF; PBac{WH}$^{f01922}$ P{XP}$^{d07256}$ /+.* This recombination deletes the *RpL28, RpL18,* and *RpL14* genes; Note the small eye size. (M) *y w eyF; PBac{WH}$^{f05041}$ P{XP}$^{d02570}$ /+;* (N) *y w eyF; PBac{WH}$^{f05041}$ P{XP}$^{d02813}$ /+.* This recombination deletes the *RpL18* gene; (O) *y w eyF; PBac{WH}$^{f05041}$ P{XP}$^{d07256}$ /+.* This recombination deletes the *RpL18* and *RpL14* genes; (P) *y w eyF; P{XP}$^{d06796}$ PBac{WH}$^{f04937}$ /+.* This recombination deletes the *eIF2γ* gene. (Q) *y w eyF; P{XP}$^{d06796}$ PBac{WH}$^{f00971}$ /+.* This recombination deletes the *eIF2γ* gene. Note the particularly small size of the recombinant heads, which also retain an unusual amount of unrecombined cells; (R) *y w eyF; P{XP}$^{d06796}$ PBac{WH}$^{f03502}$ /+.* This recombination deletes the *eIF2γ, Xrp1, RpS20* and *RpS30* genes. (S) *y w eyF; P{XP}$^{d06928}$ PBac{WH}$^{f03502}$ /+.* This recombination deletes the *Xrp1, RpS20* and *RpS30* genes. (T) *y w eyF; P{XP}$^{d06928}$ PBac{WH}$^{f01700}$ /+.* This recombination deletes the *Xrp1, RpS20,* and *RpS30* genes. (U) *y w eyF; PBac {WH}$^{f04180}$ P{XP}$^{d07944}$ /+.* The whole eye resembles the parental genotype lacking eyFlp; (V) *y w eyF; P{XP}$^{d06796}$ PBac{RB}$^{e03186}$ /+.* The whole eye resembles the parental genotype lacking eyFlp; (W) *y w eyF; P{XP}$^{d06796}$ PBac{RB}$^{e03144}$ /+.* The whole eye resembles the parental genotype lacking eyFlp. (X) *y w eyF; P{XP}$^{d08241}$ PBac{RB}$^{e02272}$ /+.* The eye has a mottled appearance indicative of small recombinant clones generated late in development; (Y) *y w eyF; P{XP}$^{d08241}$ PBac{RB}$^{e03937}$ /+.* The eye has a mottled appearance indicative of small recombinant clones generated late in development; (Z) *y w eyF; P{XP}$^{d06928}$ PBac{RB}$^{e03144}$ /+.* The eye has a mottled appearance indicative of small recombinant clones generated late in development.

larvae (**Figure 1Q,R**). Their frequency increased ~three- to fourfold in *rpS12$^{G97D}$* mutants or *p53 rpS12$^{G97D}$* double mutants (**Figure 1Q**). In the absence of irradiation, only 2 *Rp$^{+/-}$*-like bristles were observed from 500 unirradiated *rpS12$^{G97D}$* mutant flies. Since 16 macrochaetae were examined on each fly thorax, this indicated a frequency of ~1/8000 macrochaetae progenitor cells was *Rp$^{+/-}$*-like. We found none in 1000 unirradiated wild type flies (16,000 macrochaetae examined).

While the actual nature of radiation-induced genetic changes in cells forming Minute-like bristles is not directly demonstrated, previous studies of DNA-repair mutants including *mei-41*, the *Drosophila* ATR homolog, demonstrated using multiply-marked chromosomes that the majority of Minute-like bristles reflect loss of heterozygosity for large, contiguous chromosome regions (**Baker et al., 1978**). In a small-scale experiment to compare γ-irradiation to the DNA repair defects studied previously (**Baker et al., 1978**), *y$^{+/-}$ rpS12$^{G97D}$* larvae were irradiated (1000 Rad) and 2178 adult flies examined for phenotypically *y* thoracic bristles representing cells where the *y$^+$* allele had been mutated or deleted. Six times more *y* bristles were recovered in females than in males. Since the *y* locus is X-linked, this is most easily explained if *y* bristles generally result from deletions including essential genes linked to *y* that could not survive in males. In females, 37.5% of *y* bristles were also phenotypically Minute, consistent with loss of chromosome regions extending at least from the *y* locus to the *RpL36* gene 0.3 Mb more centromere-proximal that is the nearest *Rp* locus (**Figure 1S**). Although this study was small scale, these findings support the conclusion from DNA repair mutant studies that Minute-like bristles seen following irradiation most commonly reflect loss of substantial chromosome segments including *Rp* genes, (**Baker et al., 1978**). Accordingly, many Minute-like bristles removed by cell competition genes following irradiation could represent such segmentally aneuploid cells.

Taken together, these bristle results are also consistent with the notion that cell competition removes ~3/4 of the cells with genetic changes that encompass dose-sensitive *Rp* loci that arise after irradiation.

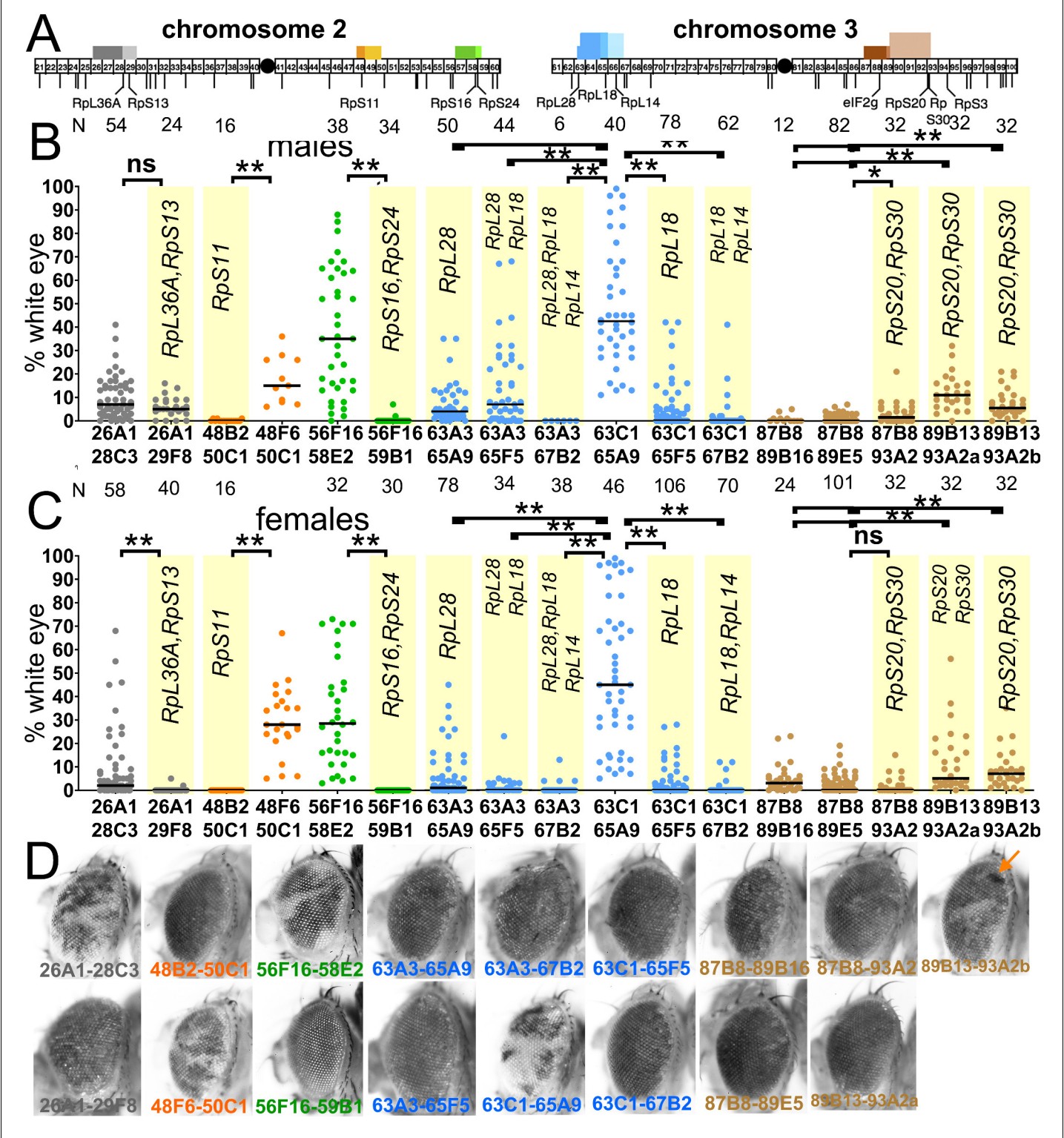

**Figure 3.** Segmental-aneuploid clones in the eye. (**A**) Cartoon of the major autosomes that contain chromosome band intervals 21–100 out of 102 total. Lines below indicate the position of all the *Rp* gene loci thought to be haploinsufficient (***Marygold et al., 2007***), as well as the *eIF2γ* locus. Identities are shown only for the loci discussed in this paper. Blocks above the chromosomes show the extent of 17 segmentally aneuploid deletions reported in this figure, color-coded according to genomic region. (**B**) Graphs show the % of the male adult eye comprising *w⁻*, segmentally aneuploid cells for 17 chromosome regions. Each data-point represents the extent of *w⁻* territory of an individual eye, median values shown as black bars, N as indicated for each genotype. Genotypes from overlapping genomic regions are shown in a common color. Yellow background indicates deletions that

*Figure 3 continued on next page*

Figure 3 continued

encompass one or more *Rp* loci. Deletion of *Rp* loci reduces contribution to the adult eye for all regions except the 87–93 region (brown). (C) As for panel A except data from females is shown. (D) Examples of typical eyes for each of the genotypes analyzed (males shown). Arrow on the Df(3R)89B13-93A2.2/+ eye indicates a small clone of darker cells reflecting four $w^+$ elements associated with tandem duplication of the 89B13-93A2 region due to a FLP-*FRT* recombination event in G2 that was not resolved by further recombination (see *Figure 2* legend). Statistics. Pairwise comparisons using the Mann-Whitney procedure with the Benjamini-Hochberg correction for multiple testing (see *Supplementary file 2*). ns – difference not statistically significant (p>0.05). * - difference significant (p<0.05). ** - difference highly significant (p<0.01). Genotypes: For 26A1-28C3, y w hsF; $P\{XP\}^{d08241}$ PBac $\{WH\}^{f04888}$/+; FRT82B/+. For 26A1-29F8, y w hsF; $P\{XP\}^{d08241}$ PBac{WH}$^{f00857}$/+; FRT82B/+. For 48B2-50C1, y w hsF; $P\{XP\}^{d09761}$ PBac{WH}$^{f00157}$/+; FRT82B/+. For 48F6-50C1, y w hsF; $P\{XP\}^{d09761}$ PBac{WH}$^{f00157}$/+; FRT82B/+. For 56F16-58E2, y w hsF; $P\{XP\}^{d02302}$ PBac{WH}$^{f04349}$/+; FRT82B/+. For 56F16-59B1, y w hsF; $P\{XP\}^{d02302}$ PBac{WH}$^{f00464}$/+; FRT82B/+. For 63A3-65A9, y w hsF; PBac{WH}$^{f01922}$ $P\{XP\}^{d02570}$ /FRT82B. For 63A3-65F5, y w hsF; PBac{WH}$^{f01922}$ $P\{XP\}^{d02813}$ /FRT82B. For 63A3-67B2, y w hsF; PBac{WH}$^{f01922}$ $P\{XP\}^{d07256}$ /FRT82B. For 63C1-65A9, y w hsF; PBac{WH}$^{f05041}$ $P\{XP\}^{d02570}$ / FRT82B. For 63C1-65F5, y w hsF; PBac{WH}$^{f05041}$ $P\{XP\}^{d02813}$ /FRT82B. For 63C1-67B2, y w hsF; PBac{WH}$^{f05041}$ $P\{XP\}^{d07256}$ /FRT82B. For 87B8-89B16, y w hsF; $P\{XP\}^{d06796}$ PBac{WH}$^{f04937}$ /FRT82B. For 87B8-89E5, y w hsF; $P\{XP\}^{d06796}$ PBac{WH}$^{f00971}$ /FRT82B. For 87B8-93A2, y w hsF; $P\{XP\}^{d06796}$ PBac {WH}$^{f03502}$ /FRT82B. For 89B13-93A2a, y w hsF; $P\{XP\}^{d06928}$ PBac{WH}$^{f03502}$ / FRT82B. For 89B13-93A2b, y w hsF; $P\{XP\}^{d06928}$ PBac{WH}$^{f01700}$ /FRT82B. The online version of this article includes the following source data for figure 3:

**Source data 1.** % eye white data.

## FLP-*FRT* recombination to generate segmental aneuploidy

Having confirmed that cell competition could potentially be important for removing cells following irradiation, we sought to assess the fate of sporadic cells that lose chromosome regions, using an assay where the cell genotypes would be definitively known and the dependence on competition with normal cells could be established. We used the FLP-*FRT* site-specific recombination system (*Golic and Lindquist, 1989*) to achieve this, exploiting large collections of transgenic flies that contain *FRT* sequence insertions at distinct chromosomal locations (*Thibault et al., 2004*). FLP recombination between pairs of *FRT* elements linked in cis excises intervening sequences to make defined deletions with a single *FRT* remaining at the recombination site (*Figure 2A,B*). Insertion elements of the Exelixis collection exist in several configurations, and FLP-mediated excision from paired FRT strains in the FRT $w^+$ ... $w^+$ FRT configuration removes both the associated $w^+$ genes, so that affected cells can be identified in the adult eye by loss of pigmentation (*Figure 2A,B*). Accordingly, we assembled a collection of genetic strains containing linked pairs of appropriate *FRT* $w^+$ elements, each flanking a distinct genomic region (*Supplementary file 1*).

It was first necessary to verify FLP recombination between *FRT* sequences, and investigate any cell-autonomous effects of the resulting segmental-monosomies. We used the eyFlp transgene, which confers continuous FLP expression to the eye and head primordia during larval life, so that FLP-*FRT* recombination is expected to approach completion (*Newsome et al., 2000*). Excision should result in white adult eyes, and also reveal any cell-autonomous effect of the resulting heterozygous deletion genotype on growth or differentiation of cells contributing to the adult eye. If the recombined genotype was autonomously cell-lethal, we would expect the developing animal to lack head structures and be unable to emerge from the pupa. If FLP-*FRT* recombination did not occur (or occurred inefficiently), we would expect adult eyes expressing the parental eye color (or with only scattered white spots that recombine late in development as cell number increases).

We identified 17 paired FRT strains that were efficient FLP targets in this assay. These 17 strains were completely or substantially white-eyed in the presence of eyFlp, indicating excision between *FRT* sites in most or all eye cells. We also identified paired FRT strains that were poor substrates for Flp (*Supplementary file 1*; *Figure 2C–T*). These either retained the parental eye color in the presence of eyFlp or produce a salt and pepper pattern of very small clones (*Supplementary file 1*; *Figure 2U–Z*). No genotype tested was inviable in the presence of eyFlp, so there was no evidence that haploinsufficiency of any of the chromosome segments tested was incompatible with cell viability or severely impacted head development. Instead, most of the 17 genotypes that recombined differentiated heads of remarkably normal external appearance and morphology (*Figure 2C–T*).

Although we did not measure head size, we noticed three genotypes in which eyes and heads were obviously smaller, consistent with a reduced growth rate of the recombined genotypes. The three small eye regions were 48B2-50C1, 565F16-59B1, and 87B8-89E5 (*Figure 2E,H,Q*). Since the *Drosophila* genome is divided cytologically into 102 band intervals, each with lettered and numbered subdivisions, in this paper we refer to the chromosome $P\{XP\}^{d09761}$ pBAC{WH}$^{f00157}$, for

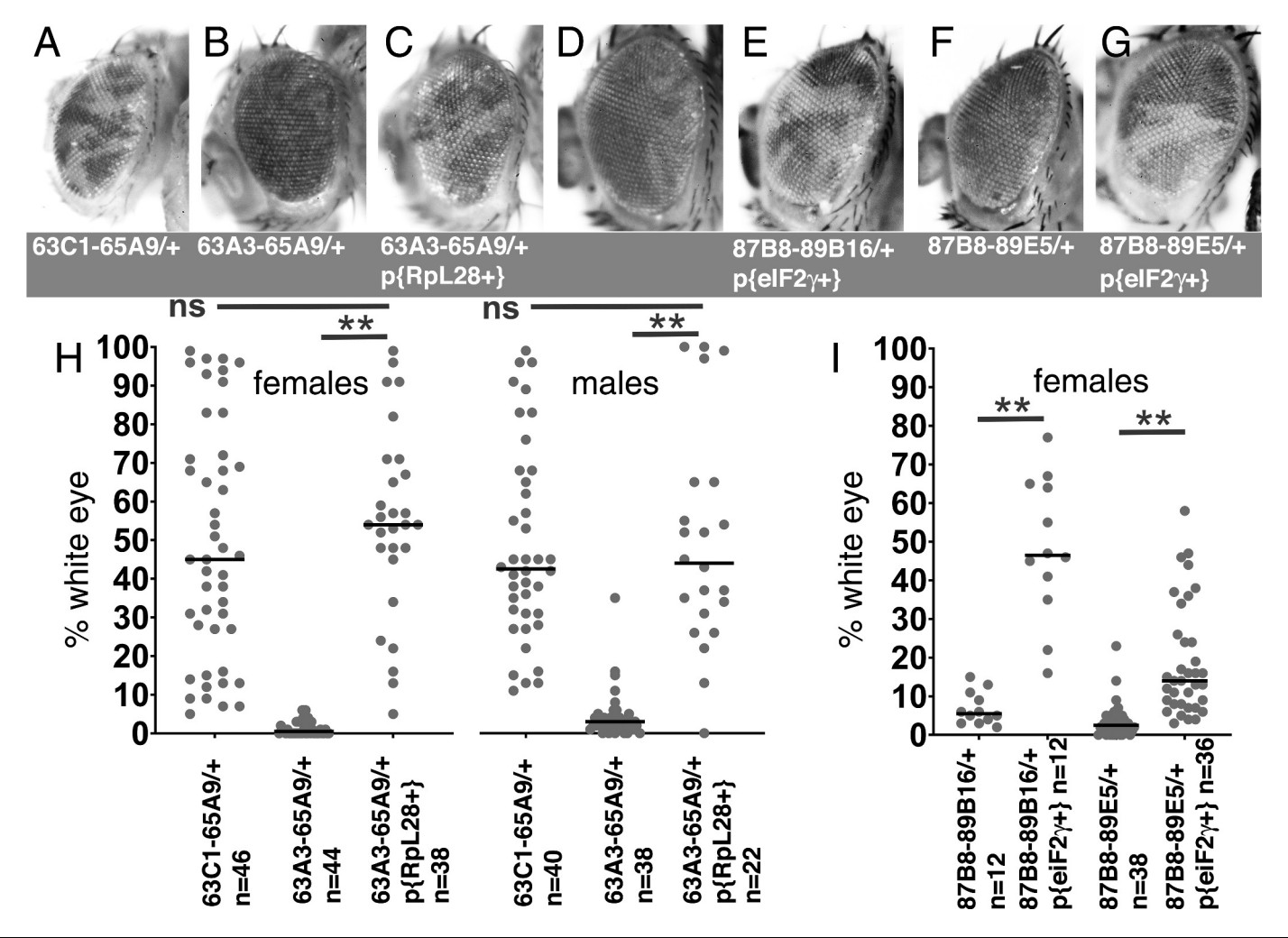

**Figure 4.** Rescue of segmental aneuploidy by single transgenes. (**A**) Df63C1-65A9/+ cells contribute ~50% of the adult eye. (**B**) Very few cells heterozygous for the overlapping Df63A3-65A9 were recovered. (**C**) Df63A3-65A9/+ cells contribute ~50% of the adult eye in the presence of the *RpL28*⁺ transgene. (**D**). Very few cells heterozygous for Df87B8-89B16 contribute to the adult eye. (**E**). Df87B8-89B16/+ cells contribute ~50% of the adult eye in the presence of the *eIF2γ* ⁺ genomic transgene. (**F**). Very few cells heterozygous for Df87B8-89E5 contribute to the adult eye. (**G**). Df87B8-89E5/+ cells contribute ~20% of the adult eye in the presence of the *eIF2γ* ⁺ genomic transgene. (**H**) Quantification of results for the *RpL28* region shown in panels A-C. Data for Df63C1-65A9 is the same as that already shown in *Figure 2A,B*. (**I**) Quantification of results for the *eIF2γ* region shown in panels D-G. Statistics. Pairwise comparisons used the Mann-Whitney procedure with the Benjamini-Hochberg correction for multiple testing (see *Supplementary file 2*). N as indicated for each genotype. ns – difference not statistically significant (p>0.05). ** - difference highly significant (p<0.01). Genotypes: For 63C1-65A9, y w hsF; *PBac{WH}^{f05041} P{XP}^{d02570}* /FRT82B. For 63A3-65A9, y w hsF; *PBac{WH}^{f01922} P{XP}^{d02570}* /FRT82B and y w hsF; *{RpL28⁺P3-DsRed}/+; PBac{WH}^{f01922} P{XP}^{d02570}* /+; FRT82B/+. For 87B8-89B16, y w hsF/+; *P{XP}^{d06796} PBac{WH}^{f04937}* /FRT82B and y w hsF/+; P{ry⁺ Su(var3-9⁺) eIF2γ⁺}/+; *P{XP}^{d06796} PBac{WH}^{f04937}* /+. For 87B8-89E5, y w hsF/+; *P{XP}^{d06796} PBac{WH}^{f00971}* /FRT82B and y w hsF/+; P{ry⁺ Su(var3-9⁺) eIF2γ⁺}/+; *P{XP}^{d06796} PBac{WH}^{f00971}* /+.

The online version of this article includes the following source data for figure 4:

**Source data 1.** % eye white data.

example, by the cytological locations of the *FRT* sequences present in the P element and PiggyBac element insertions, which are at 48B2 and 50C1 respectively in this case (*Supplementary file 1*). This nomenclature quickly communicates the genome location under study, whether it overlaps or is distinct from that affected in another strain, and also indicates that in this case the *FRT* elements are likely separated by ~2% of the genome. The full description of each insertion strain is given in *Supplementary file 1*. The three excisions that substantially reduce eye size could delete a copy of one or more haploinsufficient genes important for growth during eye development, but there also

could be a dominant effect of the novel junction generated by FLP/*FRT* recombination. Notably, excision between chromosome bands 87B8-93A2 led to eyes of normal size, although this excises all the sequences between 87B8-89E5 which led to reduced eyes. FLP recombination results in a different junctions in 87B8-89E5 and 87B8-93A2, however (*Figure 2R*).

Overall, these results showed that 17 segmentally aneuploid genotypes were cell-viable, and able to grow and differentiate in the *Drosophila* eye, although a minority might have an effect on growth in this tissue. Recombination in these 17 strains each deleted 1.4 Mb – 8.5 Mb of autosomal DNA, representing 1–6% of the sequenced genome each (*Supplementary file 1*, *Figure 3A*). Together these deletions encompass 25.3 Mb of DNA, corresponding to 21.1% of the *Drosophila* euchromatin and 17.7% of the sequenced genome. 11 of these 17 genotypes deleted one or more *Rp* loci (*RpS11*, *RpS13*, *RpS16*, *RpS20*, *RpS24*, *RpS30*, *RpL14*, *RpL18*, *RpL28* or *RpL36A*), whereas six affected no *Rp* gene (*Supplementary file 1*, *Figure 3A*). In this paper, we use the symbol *Rp* to indicate a mutation affecting any of the 66 ribosomal protein genes that are dominant through haploinsufficiency, in distinction to 13 Rp encoded by *Drosophila* loci where heterozygous mutations have no phenotype.

## *Rp* and *eIF2γ* genes determine survival and growth of segmentally aneuploid cells in mosaics

The 17 FRT pair strains that were efficient FLP targets were each exposed to a single burst of FLP expression using the heat-shock FLP transgene, intended to stimulate excision in a fraction of cells during early larval life (see Materials and methods). This led to mosaic eyes where segmentally aneuploid cells and diploid cells would be in competition. Clones of excised cells appeared only after heat-shock, confirming strict FLP-dependence.

In contrast to eyFlp recombination, mosaic eyes containing sporadic clones of excised cells were only recovered at high frequencies for four segmental aneuploid genotypes, none of which affected *Rp* loci (*Figure 3*). This confirmed that most segmentally aneuploid genotypes were selected against in mosaic eyes where diploid cells were also present, because all had been shown to be intrinsically viable when competing diploid cells were absent (*Figure 2*). Importantly, no deletion that included an *Rp* locus showed more than minimal survival of sporadic clones induced with hsFlp, suggesting *Rp* loci could be the determinants of cell competition between segmental aneuploid and wild type cells (*Figure 3B–D*). To test this in a specific case, 5.6 kb of genomic DNA encompassing the *RpL28* locus was introduced onto the second chromosome using PhiC31-mediated transgenesis. This transgene proved completely sufficient to rescue the survival of cell clones heterozygous for Df(3L)63A3-65A9, a deletion of 3.2 Mb including the *RpL28* locus (*Figure 4A–C*). Clones of Df(3L)63A3-65A9 heterozygous cells barely survived alone, with a median contribution of 2% to the eyes of males and 0% to females (*Figure 4H*). In the presence of the *RpL28*+ transgene, however, Df(3L)63A3-65A9/+ clones survived in 49 out of 50 eyes, with median contributions of 37% of the eye in males and 53% in females, not statistically different from clones heterozygous for Df(3L)63C1-65A9, an overlapping deletion of 3.0 Mb excluding the *RpL28* locus (*Figure 4H*). Thus in this case, *RpL28* gene dose alone determined whether a segmentally aneuploid genotype affecting hundreds of genes would be eliminated in competition with diploid cells.

Most of the other 10 segmental aneuploid genotypes that deleted one or more *Rp* loci contributed to adult eyes to a very significantly lower degree that overlapping segmental aneuploidies that spared *Rp* loci (*Figure 3*). This reflected the fact that, in contrast to genotypes affecting *Rp* genes, 4/6 segmental aneuploidies that spared *Rp* loci survived in eye clones at high frequencies and large sizes (*Figure 3A–C*). The two exceptions were Df(3R)87B8-89B16/+ and Df(3R)87B8-89E5/+, for which little eye tissue was recovered (*Figure 3A–C*). Although neither affected any *Rp* gene, both deleted a locus mapping to 88E5-6 encoding the translation factor eIF2γ. Since independent studies in our laboratory already identified a role for the eIF2α protein in cell competition (Kiparaki, Khan, Chuen and Baker, in preparation), we tested the possible role of *eIF2γ* by restoring *eIF2γ* diploidy to Df87B8-89B16/+ or Df87B8-89E5/+ cells using a 11.5 kb genomic transgene including the *eIF2γ* locus (*Tschiersch et al., 1994*). This completely rescued the growth and differentiation of these cells to the levels typical for aneuploidies not affecting *Rp* genes (*Figure 4D–G,I*). Thus, the locus encoding the translation factor eIF2γ behaved similarly to an *Rp* gene in triggering competitive elimination of heterozygous cells.

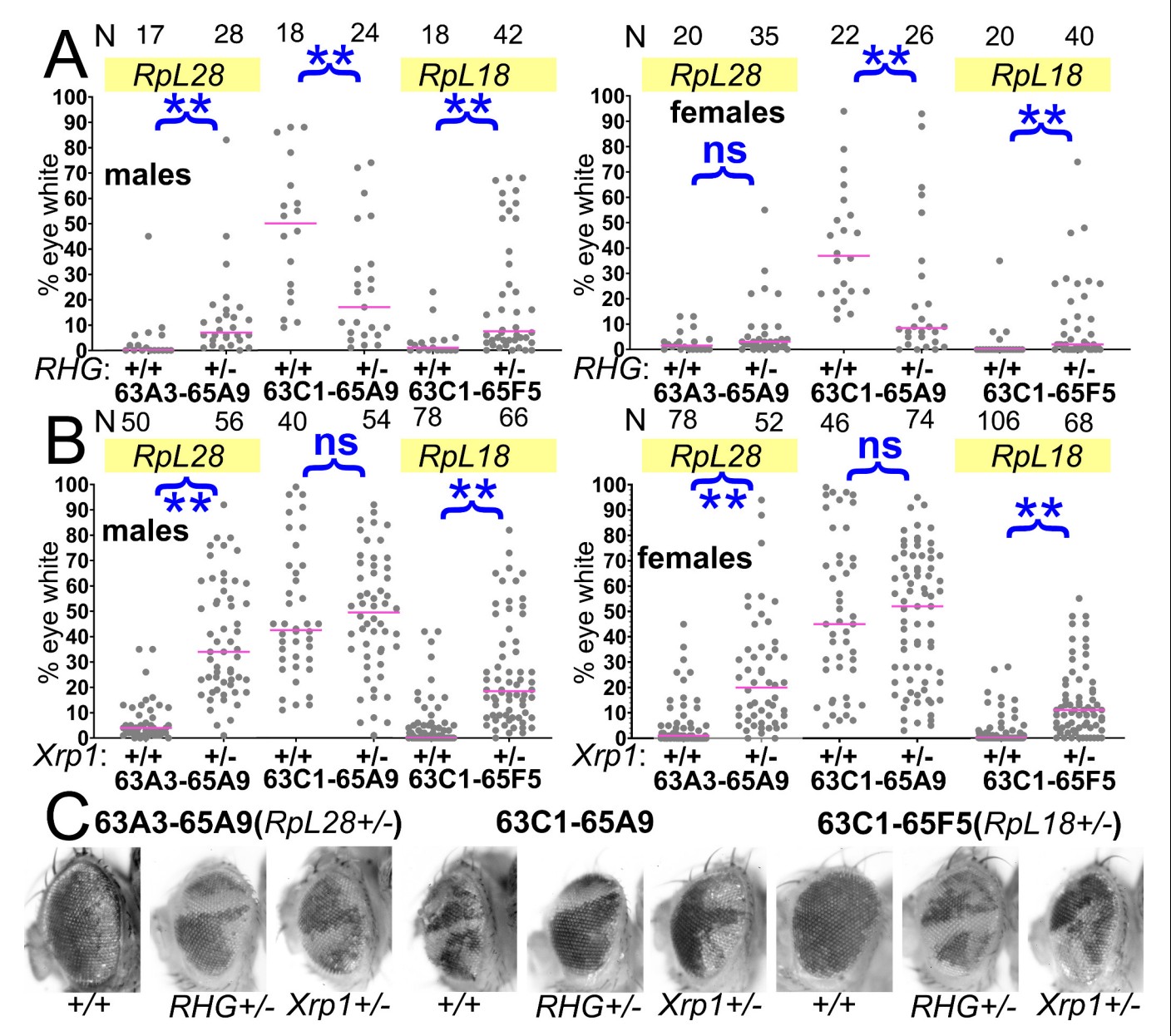

**Figure 5.** Contributions of apoptosis and Xrp1 to elimination of segmentally aneuploid cells. Panels A-B show the contribution of *w⁻* segmentally aneuploid cells to adult eyes of indicated genotypes, N as indicated for each genotype. (**A**) Heterozygosity for Df(3L)H99, which deletes the proapoptotic genes *rpr*, *hid*, and *grim* ('RHG') significantly increases contribution of Df63A3-65A9/+ and Df63C1-65F5/+ cells to adult eyes. These genotypes approach the contribution level of Df63C1-65A9/Df(3L)H99 cells which do not affect any *Rp* locus the results for 63A3-65A9/Df(3L)H99, 63C1-65A9/Df(3L)H99, and 63C1-65F5/Df(3L)H99 cannot be distinguished in males: KruskalWallis test, p=0.18; in females, this hypothesis is rejected and post-hoc testing identified the 63C1-65A9/Df(3L)H99 data as different from the other genotypes (p=0.018 for the comparison to 63A3-65A9/H99, p=0.027 for the comparison to 63C1-65F5/H99). (**B**) Heterozygosity for Xrp1, which is required for the slow growth and cell competition of *Rp⁺/⁻* point-mutant cells, significantly increases contribution of Df63A3-65A9/+ and Df63C1-65F5/+ cells to adult eyes. Xrp1 did not affect the contribution of Df63C1-65A9/+ cells that do not affect any *Rp* locus. We also tested the hypothesis that Df(3L)H99, Xrp1 mutation, and an RpS3/+ genetic background (see *Figure 8*) suppressed cell competition to an equal degree. This hypothesis was rejected for both males and females of Df63A3-65A9/+ and Df(3L)63C1-65F5/+ (Kruskal Wallis test p=$2.9 \times 10^{-17}$, $9.7 \times 10^{-21}$, $2.1 \times 10^{-5}$, and $1.8 \times 10^{-7}$, respectively). For each deletion, all the genotypes were individually significantly different except Df(3L)63C1-65F5/H99 and Df(3L)63C1-65F5/Xrp1 males, which were not significantly different (p=0.06, Conover post-hoc test with Holm correction). (**C**) Representative examples of these genotypes. Statistics. Pairwise comparisons using the Mann-Whitney procedure with the Benjamini-Hochberg correction for multiple testing (see *Supplementary file 2*). ns – difference not statistically significant (p>0.05). ** - difference highly significant (p<0.01). Multiple comparisons using the Kruskal Wallis test as described for panel B. Genotypes. For 63A3-65A9 in panel A: y w hsF; PBac{WH}^{f01922} P{XP}^{d02570} /FRT80B and y w hsF; PBac{WH}^{f01922} P{XP}^{d02570} /Df(3L)H99 FRT80B. For 63C1-65A9 in panel A: y w hsF; PBac{WH}^{f05041} P

*Figure 5 continued on next page*

Figure 5 continued

{XP}$^{d02570}$ /FRT80B and y w hsF; PBac{WH}$^{f05041}$ P{XP}$^{d02570}$ /Df(3L)H99 FRT80B. For 63C1-65F5 in panel A: y w hsF; PBac{WH}$^{f05041}$ P{XP}$^{d02813}$ /FRT80B and y w hsF; PBac{WH}$^{f05041}$ P{XP}$^{d02813}$ /Df(3L)H99 FRT80B. For 63A3-65A9 in panel B: y w hsF; PBac{WH}$^{f01922}$ P{XP}$^{d02570}$ /FRT82B and y w hsF; PBac {WH}$^{f01922}$ P{XP}$^{d02570}$ /FRT82B Xrp1$^{m2-73}$. For 63C1-65A9 in panel B: y w hsF; PBac{WH}$^{f05041}$ P{XP}$^{d02570}$ /FRT82B and y w hsF; PBac{WH}$^{f05041}$ P {XP}$^{d02570}$ /FRT82B Xrp1$^{m2-73}$. For 63C1-65F5 in panel B: y w hsF; PBac{WH}$^{f05041}$ P{XP}$^{d02813}$ /FRT82B and y w hsF; PBac{WH}$^{f05041}$ P{XP}$^{d02813}$ /FRT82B Xrp1$^{m2-73}$.

The online version of this article includes the following source data for figure 5:

**Source data 1.** % eye white data.

If these studies, which tested a significant fraction of the *Drosophila* genome, are representative, they indicate that the normal diploid complement of *Rp* loci is important for sporadic segmentally aneuploid cells to evade cell competition, and that few other genes are comparably important. The one example of such another gene uncovered in our analysis encoded eIF2γ, another protein affecting translation.

## Apoptotic genes contribute to removing segmentally aneuploid cells

If the segmentally aneuploid cells were competed by virtue of their $Rp^{+/-}$ genotypes, the genetic pathways should be similar. Elimination of $Rp^{+/-}$ point-mutant cells by competition depends on apoptosis and is suppressed by a genetic deletion, Df(3L)H99, that removes three pro-apoptotic genes *reaper* (*rpr*), *grim*, and *head-involution defective* (*hid*)(*Moreno et al., 2002*; *Kale et al., 2015*). These genes are also required for the p53-independent cell death that follows irradiation, much of which resembles cell competition (*Figure 1*; *McNamee and Brodsky, 2009*). Using the Df(3L)63A3-65A9, where clone loss was demonstrably due to heterozygosity for the *RpL28* gene (*Figure 4H*), we found that recovery of Df(3L)63A3-65A9/+ cell clones was enhanced by genetic suppression of apoptosis in the *Df(3L)H99*/+ background that experiences loss of heterozygosity for *rpr*, *grim*, and *hid* (*Figure 5A,D*). A similar rescue was obtained with the Df(3L)63C1-65F5, which deletes the *RpL18* locus (*Figure 5A,D*). The recoveries of Df(3L)63A3-65A9/Df(3L)H99 clones and Df(3L)63C1-65F5/Df (3L)H99 clones approached that of the overlapping genotype Df(3L)63C1-65A5/Df(3L)H99, in which no *Rp* genes were affected (*Figure 5A,D*). Recovery was quantitatively inferior to that seen for Df (3L)63A3-65A9/+ p{RpL28$^+$} clones (*Figure 4H*), but it is to be noted that the *Df(3L)H99*/+ background unexpectedly reduced recovery of the control Df(3L)63C1-65A5/+ cells (*Figure 5A*). Regardless of whether this reflects the recently described role for basal caspase activity in promoting imaginal disc growth in the wild type (*Shinoda et al., 2019*), or some other genetic interaction, it complicates assessment of whether H99 heterozygosity and *RpL28$^+$* transgenesis rescue Df(3L)63A3-65A9/+ clones equally. We attempted to prevent apoptosis more completely using the genetic background *hid$^{WRX1}$/ Df(3L)H99* in which *hid* is homozygously affected in addition to heterozygosity for *rpr* and *grim*. Although *hid$^{WRX1}$/ Df(3L)H99* adult animals were recovered in the absence of other mutations, they became exceptionally rare in heat-shocked combinations with the dual *FRT* chromosomes: as a result, insufficient data could be obtained to address this question. In any case, it is clear from our results that pro-apoptotic genes contribute significantly to eliminating segmentally aneuploid cells (*Figure 5A*).

## Xrp1 and RpS12 participate in removing segmentally aneuploid cells

Mutations in the *rpS12* and *Xrp1* genes are more specific for cell competition than mutations in cell death genes. The *rpS12* and *Xrp1* mutations prevent the elimination of $Rp^{+/-}$ point mutant cells from mosaics, but otherwise lead to seemingly normal flies, and do not affect other cell death processes (*Kale et al., 2018*; *Lee et al., 2018*). These genes should be required if segmental aneuploid cells are competed due to reduced *Rp* gene dose. Our results strongly support this conclusion in nearly all cases. Because the results are too extensive to present together in a single figure, they are presented in groups according to chromosome region (*Figure 5*, *Figure 6*, *Figure 7*, *Figure 8*).

Beginning with the Df(3L)63A3-65A9/+ genotype where clone loss was demonstrably due to heterozygosity for the *RpL28* gene (*Figure 4H*), we found that heterozygosity for an *Xrp1* mutation greatly restored contribution of Df(3L)63A3-65A9/+ clones (RpL28$^{+/-}$) to the eye (*Figure 5B,D*). Similar results were seen for the Df(3L)63C1-65F5/+ genotype that is heterozygous for *RpL18*, but *Xrp1* heterozygosity did not affect recovery of clones of the overlapping Df(3L)63C1-65A9/+ that is

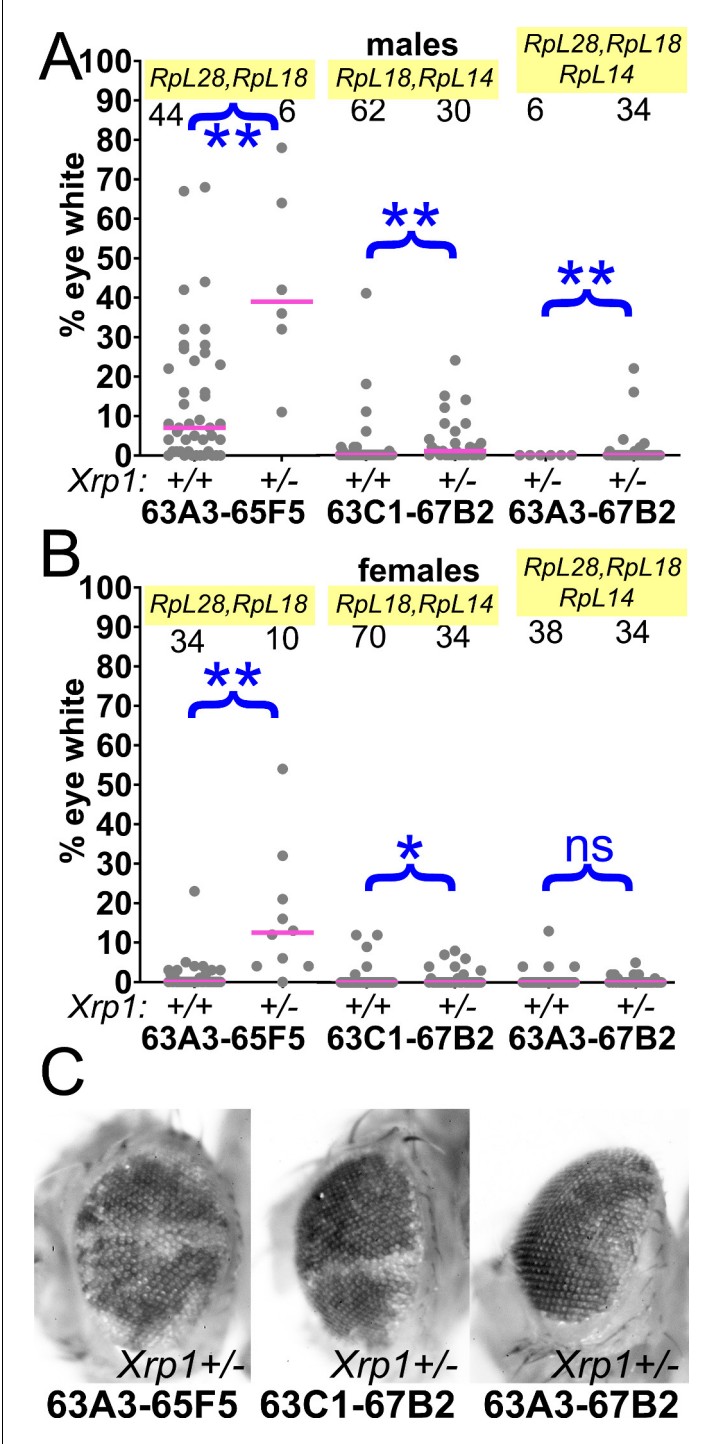

**Figure 6.** Contributions of Xrp1 to competition of larger segmental aneuploidies. (**A**) Heterozygosity for Xrp1 has progressively less effect on the contributions of larger segmental-aneuploid genotypes in males, although still significant statistically. (**B**) In females, the effect of Xrp1 on Df(3L)63A3-67B2/+ cells, haploinsufficient for *RpL14*, *RpL18*, and *RpL28*, in no longer significant statistically. *Xrp1* heterozygosity does affect Df(3L)63C1-67B2 females, although many zero values are superimposed so that this is hard to appreciate on the Df(3L)63C1-67B2/+ *Xrp1⁺/⁺* graph. (**C**). Representative examples of these phenotypes (males are shown). Statistics. Pairwise comparisons used the Mann-Whitney procedure with the Benjamini-Hochberg correction for multiple testing (see *Supplementary file 2*). N as indicated for each genotype. ns – difference not statistically significant (p>0.05). * - difference significant (p<0.05). ** - difference highly significant (p<0.01). Genotypes. For 63A3-65F5: y w hsF; *PBac*

*Figure 6 continued on next page*

*Figure 6 continued*

{WH}$^{f01922}$ P{XP}$^{d02813}$ /FRT82B and y w hsF; PBac{WH}$^{f01922}$ P{XP}$^{d02813}$ /FRT82B Xrp1$^{m2-73}$. For 63C1-67B2: y w hsF; PBac{WH}$^{f05041}$ P{XP}$^{d07256}$ /FRT82B and y w hsF; PBac{WH}$^{f05041}$ P{XP}$^{d07256}$ /FRT82B Xrp1$^{m2-73}$. For 63A3-67B2: y w hsF; PBac{WH}$^{f01922}$ P{XP}$^{d07256}$ /FRT82B and y w hsF; PBac{WH}$^{f01922}$ P{XP}$^{d07256}$ /FRT82B Xrp1$^{m2-73}$.

The online version of this article includes the following source data for figure 6:

**Source data 1.** % eye white data.

Rp$^{+/+}$ (*Figure 5B,D*). The *Xrp1* mutation even improved the survival of larger segmental-aneuploidies where combinations of the *RpL18*, *RpL28*, and *RpL14* genes were affected (*Figure 6A–C*).

Because the *rpS12*$^{G97D}$ mutation that affects cell competition recessively maps to the third chromosome, it was simpler to examine in combination with segmental aneuploidies affecting chromosome 2. These aneuploid eye clones, which like those discussed above were also recovered in the presence of an *Xrp1* mutation, included Df(2L)26A1-29F8/+, heterozygous for *RpL36A* and *RpS13*, and Df(2R)56F16-59B1/+, heterozygous for *RpS16* and *RpS24* (*Figure 7A,B,D*). Clones of the Df(2L)26A1-28C3/+ or Df(2R)56F16-58E2/+ cells that did not affect any *Rp* loci were recovered at high rates, independently of *Xrp1* genotype (*Figure 7A,B,D*). As expected, *rpS12*$^{G97D}$ homozygosity also led to significant recovery of Df(2R)56F16-59B1/+ clones that were heterozygous for *RpS16* and *RpS24*, although to a quantitatively lesser degree than *Xrp1* (*Figure 7C,D*) The contribution of Df(2R)56F16-58E2/+ cells, where no *Rp* gene is affected, was unaltered by the *rpS12*$^{G97D}$ mutation (*Figure 7C,D*).

*Xrp1* also affected other segmentally aneuploid regions. *Xrp1* mutations had a minor but statistically significant effect on clones of Df(2R)48B2-50C1/+ cells, heterozygous for *RpS11* (*Figure 8A,D*), a genotype that also had an autonomous effect on eye growth (*Figure 2E*). *Xrp1* mutations enhanced the contributions of Df(3R)87B8-89B16/+ and Df(3R)87B8-89E5/+ cells that were heterozygous for the *eIF2γ* gene (*Figure 8B–D*). Crossing to *Xrp1* only slightly improved survival of Df(3R)87B8-93A2/+ clones that were heterozygous for the *eIF2γ*, *RpS20* and *RpS30* genes, and did little to enhance recovery of Df(3R)89B13-93A2/+ genotypes that were heterozygous for *RpS20* and *RpS30* alone. As these deficiencies already delete the *Xrp1* locus itself (within 91D3-5 region), introducing an *Xrp1* mutation in trans leads to *Xrp1*$^{-/-}$ genotypes (*Figure 8B–D*). Heterozygous mutation of *Xrp1* is already sufficient to suppress competition of *Rp*$^{+/-}$ point mutant cells (*Lee et al., 2018*), probably explaining why *Xrp1* homozygosity had little further effect.

## Surrounding *Rp*$^{+/+}$ cells are necessary to eliminate segmentally aneuploid cells

Further evidence that segmentally aneuploid cells are eliminated by cell competition due to their *Rp/+* genotypes came from studies in homotypic *Rp* mutant backgrounds (*Figure 9*). It is known from previous work that cells heterozygous at two *Rp* loci do not suffer more severe competition than cells heterozygous for only one *Rp* mutation, and therefore that cells heterozygous for two *Rp* loci generally cannot be eliminated by cells heterozygous at only one *Rp* locus (*Simpson and Morata, 1981*). Accordingly, clones of segmental aneuploid cells affecting the *RpS11*, *RpS13*, *RpS16*, *RpS20*, *RpS24*, *RpS30*, *RpL14*, *RpL18*, *RpL28*, or *RpL36A* genes were all recovered significantly better in an *RpS3* point mutant background, ie *RpS3*$^{+/-}$ Df(Rp)/+ clones were not eliminated from *RpS3*$^{+/-}$ tissues (*Figure 9A,B,D*). This applied to segmental aneuploid clones heterozygous for *eIF2γ* as well (*Figure 9C,D*). On the other hand, the *RpS3* point mutant background usually had no effect on the survival of clones of genotypes that did not delete other *Rp* loci (*Figure 9C*). One exception was Df(2L)26A1-28C3/+, for which the *RpS3*$^{+/-}$ background generated significantly larger clones (*Figure 9C,D*). Notably, Df(2L)26A1-28C3/+ cells had previously been recovered less than the other non-*Rp* segmental aneuploidies (*Figure 3B,C*; *Figure 9C*). Although this could also have reflected a lower rate of FLP-recombination between the 26A1 and 28C3 *FRT* sites, in the *RpS3*$^{+/-}$ background the recovery of Df2(2L)6A1-28C3/+ clones was similar to that of Df(2R)48F6-50C1/+, Df(2R)56F16-58E2/+ or Df(3L)63C1-65A9/+, suggesting instead that Df(2L)26A1-28C3/+ might be subject to a mild cell competition that can be rescued in the *RpS3*$^{+/-}$ background (*Figure 9C*).

Although suppression of apoptosis, mutation of cell competition genes, or a germline-inherited *RpS3* background all restored the growth and survival of cells hemizygous for *Rp* loci, they may not

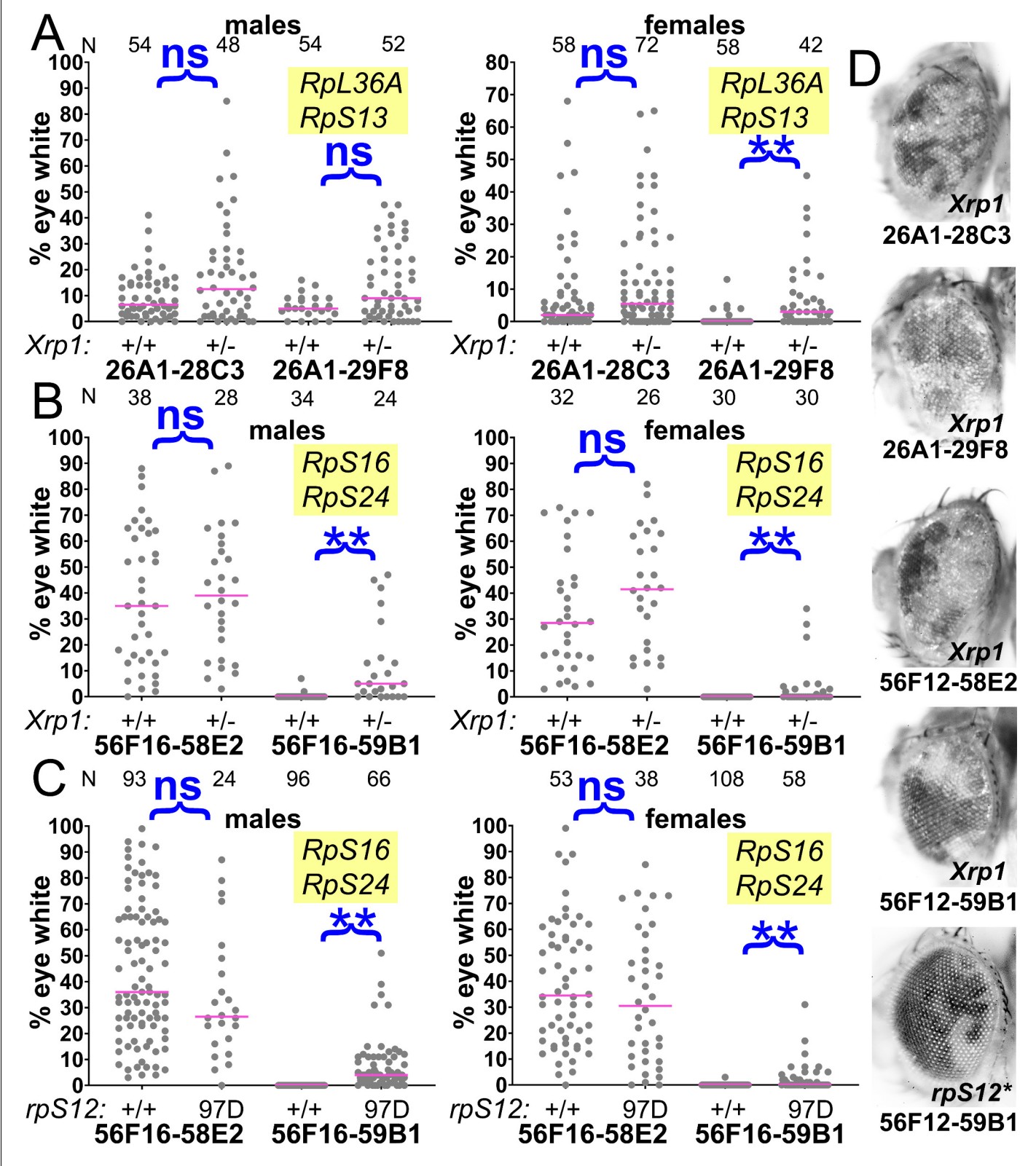

**Figure 7.** Contributions of Xrp1 and RpS12 to competition of chromosome two genotypes. (**A**) Xrp1 mutation improved the eye contributions of the Df(26A1-29F8)/+ genotype. Data for Df(26A1-28C3)/+ are the same as shown in *Figure 2*. (**B**) Xrp1 mutation improved the eye contribution of Df(56F16-59B1)/+, heterozygous for *RpS16* and *RpS24*, but had no effect on the contribution of Df(56F16-58E2)/+ cells that affect no *Rp* loci. (**C**) Homozygosity for the *rpS12^{G97D}* mutation improved the eye contribution of Df(56F16-59B1)/+, heterozygous for *RpS16* and *RpS24*, but had no effect on the

*Figure 7 continued on next page*

Figure 7 continued

contribution of Df(56F16-58E2)/+ cells that affect no *Rp* loci. (D) Representative examples of these phenotypes (males shown). Statistics. Pairwise comparisons used the Mann-Whitney procedure with the Benjamini-Hochberg correction for multiple testing (see *Supplementary file 2*). N as indicated for each genotype. ns – difference not statistically significant (p>0.05). ** - difference highly significant (p<0.01). Genotypes. For 26A1-28C3: y w hsF; P{XP}$^{d08241}$ PBac{WH}$^{f04888}$/+; FRT82B/+ and y w hsF; P{XP}$^{d08241}$ PBac{WH}$^{f04888}$/+; FRT82B Xrp1$^{m2-73}$/+. For 26A1-29F8: y w hsF; P{XP}$^{d08241}$ PBac{WH}$^{f00857}$/+; FRT82B/+ and y w hsF; P{XP}$^{d08241}$ PBac{WH}$^{f00857}$/+; FRT82B Xrp1$^{m2-73}$/+. For 56F16-58E2 in panel B: y w hsF; P{XP}$^{d02302}$ PBac{WH}$^{f04349}$/+; FRT82B/+ and y w hsF; P{XP}$^{d02302}$ PBac{WH}$^{f04349}$/+; FRT82B Xrp1$^{m2-73}$/+. For 56F16-59B1 in panel B: y w hsF; P{XP}$^{d02302}$ PBac{WH}$^{f00464}$/+; FRT82B/+ and y w hsF; P{XP}$^{d02302}$ PBac{WH}$^{f00464}$/+; Xrp1$^{m2-73}$/+. For 56F16-58E2 in panel C: y w hsF; P{XP}$^{d02302}$ PBac{WH}$^{f04349}$/+; FRT80B/FRT80B and y w hsF; P{XP}$^{d02302}$ PBac{WH}$^{f04349}$/+; rpS12$^{G97D}$ FRT80B/rpS12$^{G97D}$ FRT80B. For 56F16-59B1 in panel B: y w hsF; P{XP}$^{d02302}$ PBac{WH}$^{f00464}$/+; FRT80B/FRT80B and y w hsF; P{XP}$^{d02302}$ PBac{WH}$^{f00464}$/+; rpS12$^{G97D}$ FRT80B/rpS12$^{G97D}$ FRT80B.

The online version of this article includes the following source data for figure 7:

**Source data 1.** % eye white data.

have done so equally. Suppressing apoptosis was least effective at expanding the contribution of aneuploid cells in the rescued eyes (*Figure 5A*). *Xrp1*, *rpS12*$^{G97D}$, and the *RpS3*$^{+/-}$ background suppressed cell competition to similar extents, although the general *RpS3* background often had the greatest effect, comparable to those of *RpL28*$^+$ and *eIF2γ*$^+$ transgenes (see statistical comparisons for the 63A3-65F5 and 56F16-59B1 regions in *Figures 5* and *6* legends). Although several explanations could justify these differences, it is worth noting that the results correlate with the effects of these genetic backgrounds on translation and growth. Thus Df(3L)H99, which suppresses apoptosis with no known increase translation or cellular growth, is expected to suppress the competition of *Rp*$^{+/-}$ cells but not restore their translation. As a consequence, clones of *Rp*$^{+/-}$ Df(H99)/+ cells, although surviving, are not expected to grow as rapidly or contribute as much to the eye as clones of *Rp*$^{+/+}$ cells. By contrast the *rpS12*$^{G97D}$ and *Xrp1* mutations restore the general translation rate of *Rp*$^{+/-}$ cells, with *Xrp1* mutation also restoring more normal rates of cellular and organismal growth (*Lee et al., 2018*; *Ji et al., 2019*). A background mutation in *RpS3* is not expected to restore translation or growth to segmentally aneuploid cells, but by equally impairing the unrecombined cells, and systemically delaying the growth and developmental rate of the organism as a whole, it equalizes the contributions of aneuploid and control genotypes.

In summary, our results strongly support the conclusion that the growth and survival of most segmentally aneuploid regions is determined by cell competition according to *Rp* gene copy number, and show that the RpS12/Xrp1-dependent process that eliminates *Rp*$^{+/-}$ point mutated cells also acts on cells with large losses of genetic material that include *Rp* genes.

### Competition of segmentally aneuploid cells in the thorax

Experiments using the hsFLP transgene should stimulate recombination and segmental aneuploidy in all tissues, not only in the eye where excision causes loss of pigmentation. To test this, we looked for cells with deletions encompassing *Rp* loci in the thorax, where *Rp* haploinsufficiency leads to small, thin thoracic bristles (*Marygold et al., 2007*). This was explored using Df(2R)56F16-59B1, which deletes the *RpS16* and *RpS24* loci. Minute-like bristles were not observed on the thoraces of heat-shocked flies carrying Df(2R)56F16-58E2 heterozygous clones, which affect no *Rp* locus, or on the thoraces of heat-shocked 56F16-59B1 flies lacking rpS12 or *Xrp1* mutations, but they represented 0.5% of the thoracic bristles in the 56F16-59B1/+ *rpS12*$^{G97D}$ flies and 0.25% of the thoracic bristles in the 56F16-59B1/+ *Xrp1*$^{m2-73/+}$ background (*Figure 10A,B*). These findings indicate that Df(2R)56F16-59B1/+ cells also survive to adulthood in the thorax if cell competition is suppressed, albeit at lower frequency than observed in the eye.

## Discussion

We sought to test the hypothesis that cell competition is a mechanism that can target aneuploid cells based on their altered *Rp* gene dose (*McNamee and Brodsky, 2009*). It was already known that cells carrying point mutations at *Rp* loci are eliminated from developing imaginal discs by cell competition (*Morata and Ripoll, 1975*; *Simpson, 1979*; *Baker, 2020*). Here, we tested whether cells with more extensive genetic defects that reduce *Rp* gene dose also experience cell competition, and if so how significant this is for the removal of cells with damaged genomes.

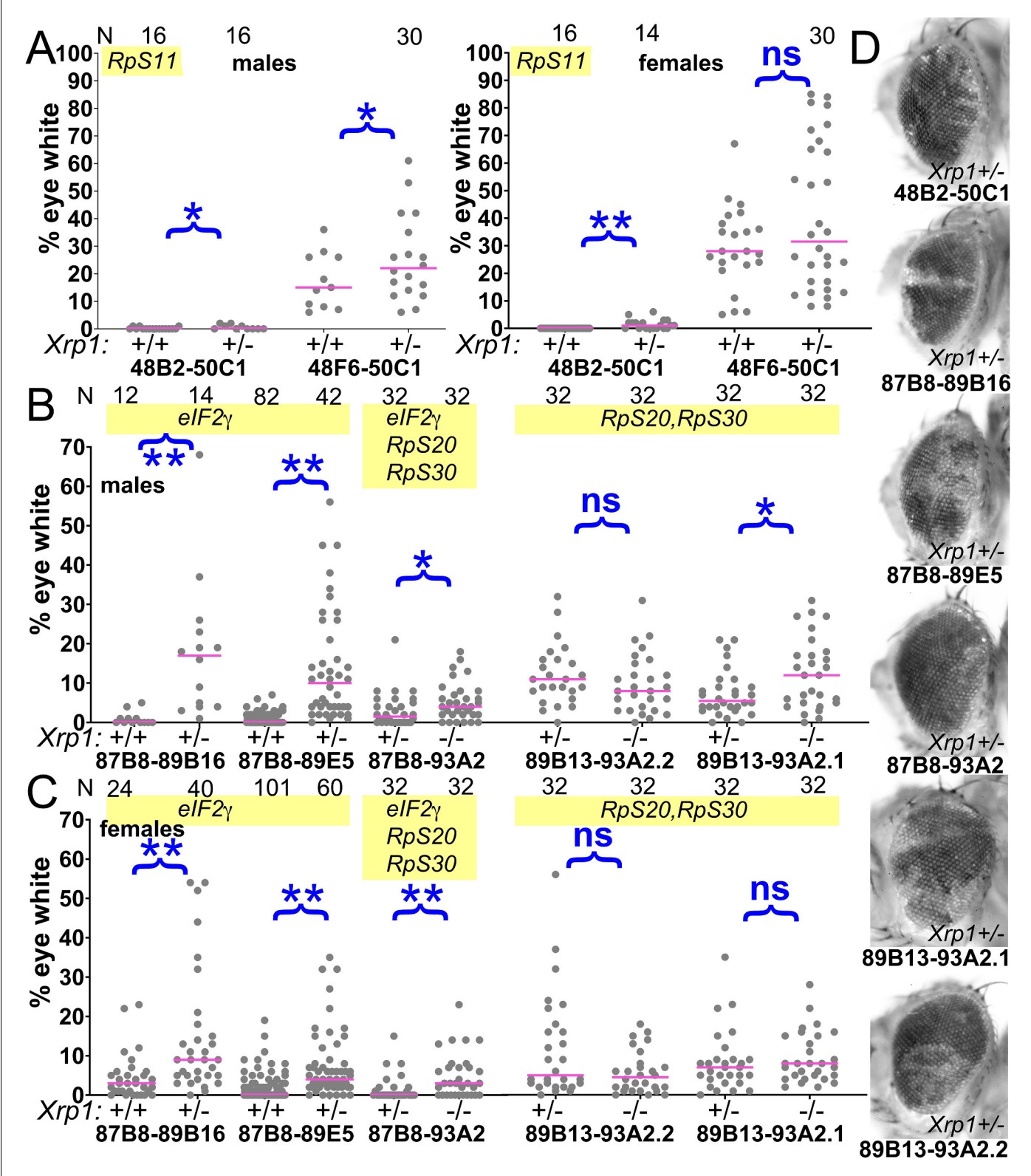

**Figure 8.** Contribution of Xrp1 to competition of further genotypes. (**A**) *Xrp1* mutation made a small but significant improvement to the contribution of Df(48B2-50C1)/+ cells, heterozygous for *RpS11*, but its effect on Df048F6-50C1/+ cells, which have no affected *Rp* genes, was only significant in males. (**B,C**) *Xrp1* mutation significantly rescued the contribution of all segmental aneuploid genotypes affecting the *eIF2γ* locus. There was little effect of mutating the second *Xrp1* copy on Df(89B13-93A2.1)/+ and Df(89B13-93A2.2)/+ cells, both of which are haplo-insufficient for *RpS20* and *RpS30* as well

*Figure 8 continued on next page*

Figure 8 continued

as *Xrp1*. (D) Representative examples of these genotypes (males shown). For panels A–C, the control data (*Xrp1*$^{+/+}$ for A and *Xrp1*$^{+/-}$ for (B,C) were also shown in *Figure 2*). Statistics. Pairwise comparisons used the Mann-Whitney procedure with the Benjamini-Hochberg correction for multiple testing (see *Supplementary file 2*). N as indicated for each genotype. ns – difference not statistically significant (p>0.05). * - difference significant (p<0.05). ** - difference highly significant (p<0.01). Genotypes. For 48B2-50C1: y w hsF; P{XP}$^{d09761}$ PBac{WH}$^{f00157}$/+; FRT82B/+ and y w hsF; P{XP}$^{d09761}$ PBac{WH}$^{f00157}$/+; FRT82B Xrp1$^{m2-73}$/+. For 48F6-50C1: y w hsF; P{XP}$^{d09761}$ PBac{WH}$^{f00157}$/+; FRT82B/+ and y w hsF; P{XP}$^{d09761}$ PBac{WH}$^{f00157}$/+; FRT82B Xrp1$^{m2-73}$/+. For 87B8-89B16: y w hsF; P{XP}$^{d06796}$ PBac{WH}$^{f04937}$/FRT82B and y w hsF; P{XP}$^{d06796}$ PBac{WH}$^{f04937}$/FRT82B Xrp1$^{m2-73}$. For 87B8-89E5: y w hsF; P{XP}$^{d06796}$ PBac{WH}$^{f00971}$/FRT82B and y w hsF; P{XP}$^{d06796}$ PBac{WH}$^{f00971}$/FRT82B Xrp1$^{m2-73}$. For 87B8-93A2: y w hsF; P{XP}$^{d06796}$ PBac{WH}$^{f03502}$/FRT82B and y w hsF; P{XP}$^{d06796}$ PBac{WH}$^{f03502}$/FRT82B Xrp1$^{m2-73}$. For 89B13-93A2.1: y w hsF; P{XP}$^{d06928}$ PBac{WH}$^{f03502}$/FRT82B and y w hsF; P{XP}$^{d06928}$ PBac{WH}$^{f03502}$/FRT82B Xrp1$^{m2-73}$. For 89B13-93A2.2: y w hsF; P{XP}$^{d06928}$ PBac{WH}$^{f01700}$/FRT82B and y w hsF; P{XP}$^{d06928}$ PBac{WH}$^{f01700}$/FRT82B Xrp1$^{m2-73}$.

The online version of this article includes the following source data for figure 8:

**Source data 1.** % eye white data.

The idea that cell competition eliminates aneuploid cells developed from studies of cellular responses to DNA damage, where a delayed, p53-independent process follows after the acute, p53-dependent DNA damage response (*Wichmann et al., 2006*; *Titen and Golic, 2008*; *McNamee and Brodsky, 2009*). We found that a substantial proportion of p53-independent cell death shared genetic requirements with cell competition, consistent with cell competition being responsible (*Figure 1*). Accordingly, when the cell competition pathway was inhibited, more Minute-like bristles were recovered on the irradiated flies, an indication that cell competition could be removing cells that experience substantial losses of genetic material (*Figure 1R*). A proportion of both the p53-independent cell death and Minute-like bristles were independent of *rpS12*, however, suggesting that cell competition might not be the only process at work.

To measure the role of cell competition on defined genotypes, where the role of surrounding wild type cells could also be assessed, we then used site-specific recombination to excise chromosome segments from isolated cells during imaginal disc development. As expected, segmental aneuploidy prevented cells contributing clones to the adult eye whenever *Rp* gene dose was reduced (*Figure 11A*). More significantly, cell competition appears to be the primary mechanism limiting the contribution of segmental aneuploidies in the tested size ranges to adult tissues, because segmental aneuploid cells easily survived and contributed large fractions of the adult tissue when they did not affect *Rp* loci, when diploidy for *Rp* loci was restored with a transgene, or when the cell competition pathway that depends on RpS12 and Xrp1 function was mutated.

The segmental-aneuploid genotypes examined here were able to form entire heads of aneuploid cells when eyFlp was used to drive recombination in all the cells (*Figure 2*). The removal of sporadic aneuploid cells therefore depended on competition with diploid cells. In fact in the cases of Df(2R)56F16-59B1/+, heterozygous for the *RpS16* and *RpS24* genes, and Df(3L)65A5-65A9/+, heterozygous for the *RpL28* gene, we bred flies that received heat-shock recombination, and recovered non-mosaic, entirely segmentally aneuploid flies in the next generation, derived from FLP-*FRT* recombination in the germlines of the parents. Thus, these segmentally aneuploid genotypes, which rarely survived in sporadic clones, were viable in all tissues when competing wild type cells were not present.

The most effective suppression of *Rp*$^{+/-}$ segmental aneuploid clones was generally seen when the whole animal was heterozygous for a point mutation in *RpS3* (*Figure 9*). This is further, compelling evidence that cell competition due to reduced *Rp* gene dose is the main mechanism eliminating segmentally aneuploid because it shows that the feature of euploid cells that enables them to eliminate aneuploid cells is their *Rp*$^{+/+}$ genotype.

In contrast to these results, segmental aneuploidy leaving *Rp* loci unaffected was compatible with clonal growth and differentiation for four of the five genomic regions tested (*Figure 3*). In the exception, we identified *eIF2γ* as the locus responsible for loss Df(3R)87B8-89B16/+ clones and Df(3R)87B8-89E5/+ clones (*Figure 4I*). No point mutant alleles of the *eIF2γ* gene are known and the locus is believed to be haplo-lethal to *Drosophila* (*Marygold et al., 2007*). It is cell competition that eliminates *eIF2γ*$^{+/-}$ aneuploid cells from the eye, however, since they could form apparently normal adult heads when no diploid competitor cells were present (*Figure 2P–R*). Moreover, clones of the *eIF2γ*$^{+/-}$ genotypes Df(3R)87B8-89B16/+ and Df(3R)87B8-89E5/+ were restored by both the *Xrp1*

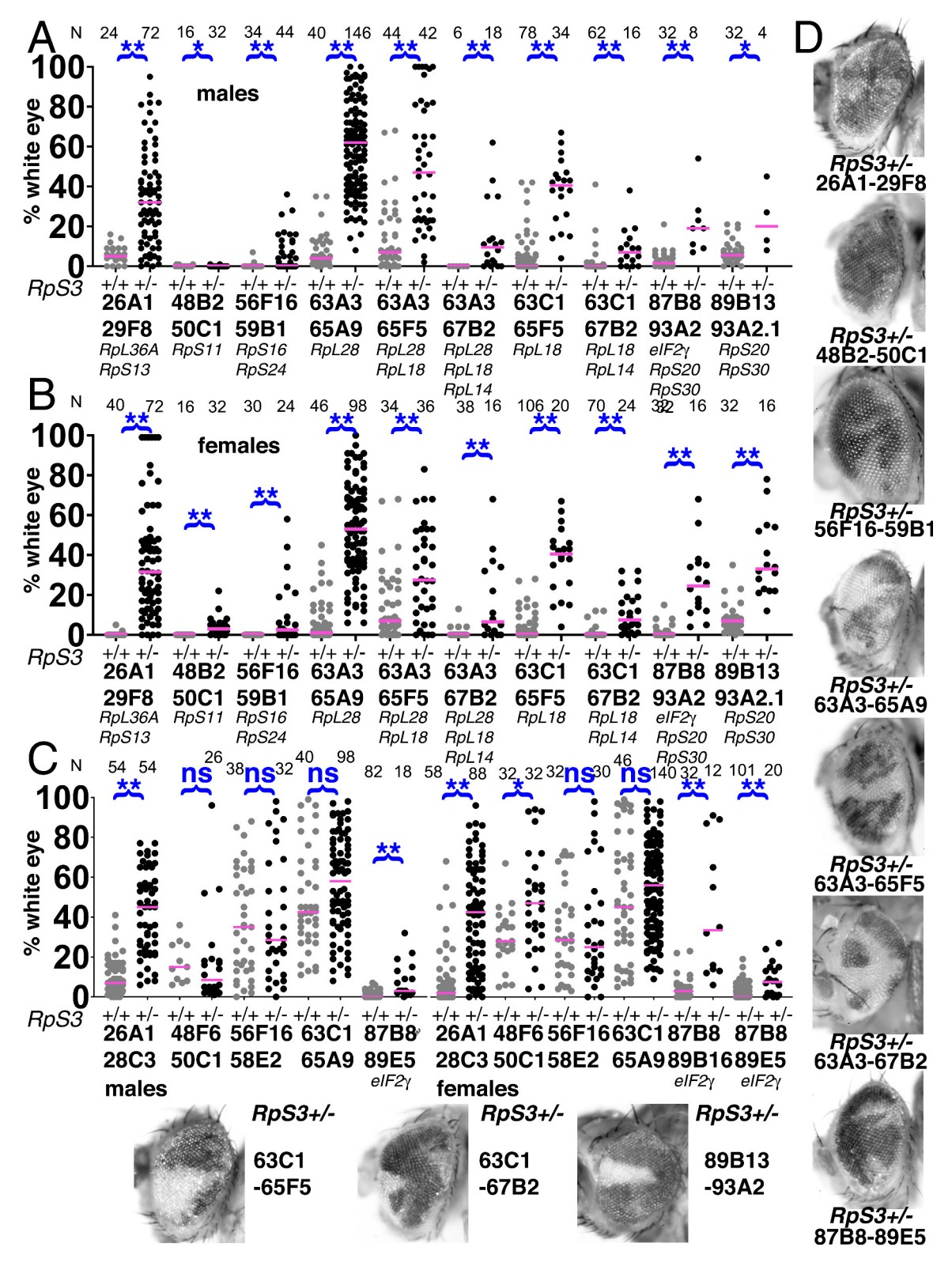

**Figure 9.** Elimination of segmentally aneuploid clones requires a $Rp^{+/+}$ background. Eye contributions of clones of segmentally aneuploid genotypes deleting the indicated $Rp$ loci. in $RpS3^{+/+}$ and $RpS3^{+/-}$ genetic backgrounds. Magenta bars show the median contributions, N as indicated for each genotype. (**A**) In males, the $RpS3^{+/-}$ genetic background (black datapoints) allows for significantly greater contribution in all cases except Df(48B2-50C1)/+. No Df(3R)89B13-93A2.2/$RpS3$ flies were obtained, this genotype is lethal due to an unidentified shared lethal outside the 89B13-93A2.2 region.

*Figure 9 continued on next page*

*Figure 9 continued*

(**B**) Comparable data from females. The *RpS3⁺/⁻* genetic background (black datapoints) always allows for significantly greater contribution. (**C**) Eye contributions of clones of segmentally aneuploid cells where no *Rp* loci are affected. The *RpS3⁺/⁻* genetic background (black datapoints) had no significant effect on many such genotypes, but did enhance the contribution of Df(26A1-28C3)/+ clones and of the Df(3R)87B8-89B16/+ and Df(3R)87B8-89E5/+ clones that were haploinsuffiicent for *eIF2γ* ⁺. No Df(3R)87B8-89B16/*RpS3* males were obtained. (**D**) Representative examples of these *RpS3⁺/⁻* genotypes (males shown). The *RpS3⁺/⁺* control data were shown previously in *Figure 2*, except for 87B8-89B16 females. Statistics. Pairwise comparisons using the Mann-Whitney procedure with the Benjamini-Hochberg correction for multiple testing (see *Supplementary file 2*). ns – difference not statistically significant (p>0.05). * - difference significant (p<0.05). ** - difference highly significant (p<0.01). Genotypes: Same as for *Figure 3*, with an *FRT82B RpS3* chromosome substituting for *FRT82B* where indicated.

The online version of this article includes the following source data for figure 9:

**Source data 1.** % eye white data.

---

mutant and by the *RpS3⁺/⁻* mutant background, as expected for cell competition (*Figure 8*). It is possible that the 26A1-28C3 region might also contain a non-*Rp* gene whose deletion leads to a cell competition, although much less severe.

Out of 63 other translation factor genes examined in a systematic study of whole body, non-mosaic phenotypes, *eIF2α* and *eIF2γ* were the only haploinsufficient loci found (*Marygold et al., 2007*). Notably, the *eIF2α* gene is the only other locus known where point mutants lead to the developmental delay and thin bristle phenotype that is otherwise typical of heterozygous *Rp* mutants (*Marygold et al., 2007*), suggesting a functional relationship between *Rp* mutants and the eIF2 complex. Independent studies in our laboratory already indicate that eIF2α is regulated by Xrp1 and contributes directly to the cell competition mechanism (Kiparaki, Khan, Cheun and Baker, in preparation).

Previous studies suggested that cells with whole chromosome aneuploidies experience a stress associated with mismatched dose of many proteins (*Torres et al., 2007*; *Zhu et al., 2018*; *Terhorst et al., 2020*). We cannot measure how such stresses reduced clonal growth of segmental

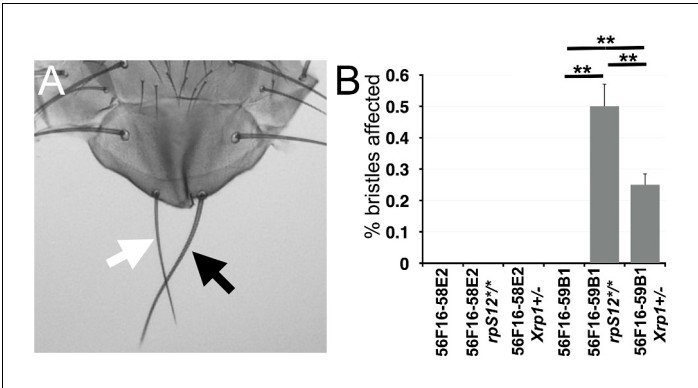

**Figure 10.** Segmentally aneuploid cells in the thorax. (**A**) White arrow indicates a 'Minute'(*Rp⁺/⁻*)-like short thin scutellar bristles on a fly containing clones of 56F16-59B1 cells. Compare the normal contralteral scutellar bristles (black arrow). (**B**) Frequency of affected bristles in genotypes as indicated. Data represent averages from three sets of 100 adults of each genotype. Minute-like bristles were not seen with the 56F16-58E2 deletion that does not affect any *Rp* locus. This provides as a baseline for any spontaneous loss of heterozygosity for *Rp* gene that might occur unrelated to the FLP-*FRT* excision, and which is evidently rare. Minute-like bristles were also not seen on cell competition-competent flies where 56F16-59B1 excisions would create heterozygosity for *RpS16* and *RpS24*. These bristles only appeared in the *rpS12* and *Xrp1* mutant backgrounds where cell competition was compromised. Statistics. Three sets of 100 flies analyzed for each genotype. One-way ANOVA rejects the hypothesis that the six datasets are indistinguishable (p=4.28×10⁻⁷). The Holm procedure for multiple comparisons showed that results for 56F16-59B1 in the *rpS12* and *Xrp1* backgrounds were different from all others and from one another (adjusted p<0.05). For simplicity, significance is only indicated for 56F16-59B1 genotypes. ns – difference not statistically significant (p>0.05). * - difference significant (p<0.05). ** - difference highly significant (p<0.01).

The online version of this article includes the following source data for figure 10:

**Source data 1.** Source data for *Figure 10B*.

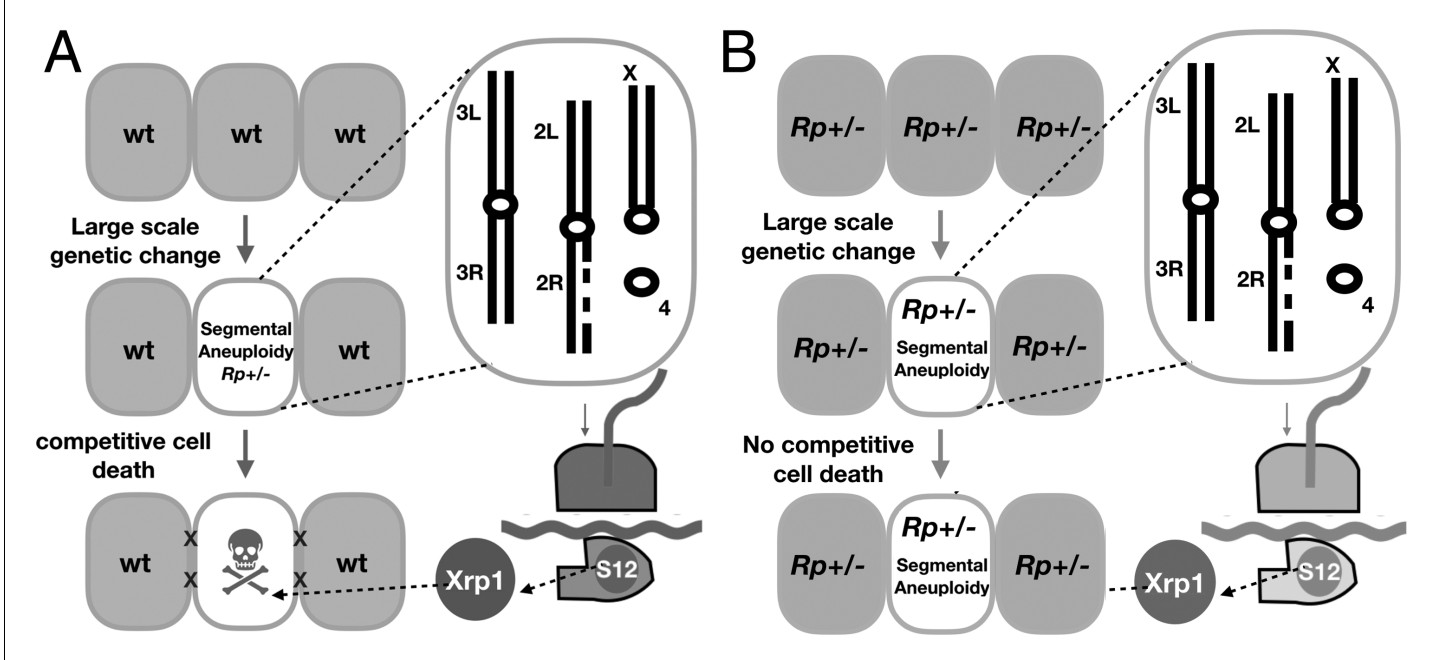

**Figure 11.** Models. (**A**) Model for the elimination of segmentally aneuploid cells from imaginal discs. Cells that lose part of one chromosome may become haploinsufficient for one or more *Rp* loci, affecting ribosome assembly and triggering RpS12 and Xrp1 activities that lead to cell elimination by competition with unaffected neighboring cells. (**B**) Our studies show that segmentally aneuploid cells proliferate and contribute to adult tissues in animals where all cells are heterozygous for an *Rp* mutation, reducing the difference between aneuploid and diploid cells. A similar situation might apply in Diamond-Blackfan patients, many of whom are haploinsufficient for *Rp* loci. If *Rp* loci are also indicators of chromosome rearrangements in mammalian cells, such patients might face a greater accumulation of aneuploid cells.

aneuploid cells in our experiments, but the effect must be small compared to cell competition, since without cell competition, segmental aneuploid cells easily contributed half or more of the eye, whereas cell competition drastically reduces this contribution. It seems unlikely that all five independent genomic regions examined here, comprising 21.1% of the euchromatic genome, represent exceptional cases. It is plausible, however, that additional stresses increase with more extensive loss of genetic material, eg clones heterozygous for a 6.3 Mb deletion removing the *RpL14*, *RpL18*, and *RpL28* loci were recovered less well than smaller deletions (*Figure 6*), as if larger monosomies experience other stresses in addition to cell competition. Since extra copies of at least two *Rp* genes (RpS12 and RpL36) do not trigger cell removal (*Kale et al., 2018*), other mechanisms would also be required to eliminate cells with triploidies, or act in tissues that lack cell competition (*Ripoll, 1980*). Finally, preventing apoptosis of *Drosophila* cells that have chromosome instability leads to invasive tumor growth that can be propagated after transplantation (*Dekanty et al., 2012*; *Benhra et al., 2018*). We did not observe invasive growth after blocking cell death of segmentally aneuploid cells, suggesting that chromosome instability may lead to different classes of aneuploidy, or to other additional effects.

If cell competition is the main mechanism eliminating cells with segmental monosomies, at least up to a certain size, how important is this? The segmental aneuploidies we studied were comparable in genetic content to some whole chromosome monosomies in humans. For example, cells heterozygous for Df(2R)56F16-58E2 were hemizygous for a 2.2 Mb region including 1.5% of the genome, about as large a region as can be expected to lack any *Rp* gene, encoding 333 protein coding genes and 55 non-coding RNAs. Human chromosome 21, which also contains 1.5% of the genome that lacks any *Rp* gene, encodes 234 protein coding genes and 404 non-coding RNAs (*Uechi et al., 2001*). The similarity is not coincidental, because Rp number is conserved and the total gene number is also comparable, so genome segments that lack *Rp* loci are expected to be similar when measured by gene number or fraction of the genome. Thus, Df(2R)56F16-58E2 is comparable in genetic terms to loss of a small human chromosome. Some of the segmental aneuploidies we studied in

*Drosophila* were several-fold larger than Df(2R)56F16-58E2 (**Supplementary file 1**; **Figure 3A**), Thus, our studies may best model aneuploidies affecting one or a few human chromosomes. Because ~80% of the *Drosophila* genome is carried on two autosomes, whole-chromosome aneuploidies in *Drosophila*, by contrast, better mimic complex karyotypes seen in tumors or in cells with chromosome instability, which affect many chromosomes.

Little is known about what aneuploidies arise spontaneously in normal development. Ionizing radiation generates many kinds of chromosome aberration, so if more than half the p53-independent cell death following irradiation resembles cell competition (**Figure 1**), this suggests cell competition could be significant for removing many, although not all, the damaged cells that arise.

Could cell competition be important in humans? As in *Drosophila*, the 80 *Rp* gene loci are distributed seemingly randomly around the 24 pairs of human chromosomes (**Uechi et al., 2001**). At least 21 human *Rp* genes have so far been found to be haploinsufficient and are responsible for the dominant syndrome Diamond Blackfan Anemia (DBA)(**Ulirsch et al., 2018**). Thus *Rp* genes could be sensors for aneuploidy in humans. The retention of aneuploid cells in *Drosophila* that inherit an *Rp* mutation from the germline (**Figure 9**) resembles the situation in human DBA patients, the majority of whom are constitutively heterozygous for a *Rp* gene mutation or deletion (**Ulirsch et al., 2018**). DBA patients experience 4.8x higher lifetime incidence of multiple cancers, not limited to the hematopoietic system (**Vlachos et al., 2018**). Current hypotheses for this cancer predisposition include specific alterations to the spectrum of translation due to defective ribosome biogenesis, a loss of translational fidelity due to selection of second-site suppressor mutations, selective pressure for p53 mutations due to the chronic p53 activity in such genotypes, and oxidative stress or metabolic reprogramming in $Rp^{+/-}$cells (**Sulima et al., 2019**). To these we can now add the possibility that DBA patients experience a diminished capacity to recognize and eliminate aneuploid cells, because their euploid cells are not $Rp^{+/+}$(**Figure 11B**). The nearly fivefold increase in tumor incidence suggests that if this was correct, cell competition might remove as many as 80% of pre-neoplastic cells from normal individuals due to their aneuploidy, This seems comparable to our findings that cell competition removes 58–86% of the cells with radiation-damaged genomes in *Drosophila* (**Figure 1M,P,R**).

## Materials and methods

### Fly strains and FLP-*FRT* methods

Flies were reared on standard medium at 25°C unless otherwise noted. The genetic strains used are described in the Key Resources Table.

Strains carrying pairs of *FRT* transgenic elements in cis were obtained after meiotic recombination using appropriate genetic crosses, monitoring recombination frequency to confirm the expected transgene locations. FLP expression was induced by 37°C heat shock for 30 or 60 min at 36 ± 12 hr after egg laying. Adult flies were aged ~1 week to allow eye color to darken fully, then stored at −20°C for later measurement and photography. The fraction of each adult eye populated by unpigmented cells was estimated manually under a dissecting microscope. Samples were blinded for genotype before scoring by an independent investigator. We estimate the clonal composition of the eye by conceptually dividing each eye into segments so as to focus on the composition of the mosaic subregions. For example, an eye that is 56% white might be half white with an additional quarter of the eye that was one quarter white. Estimates are no doubt approximate although we do not think the errors are large. Importantly, the Mann Whitney procedure used to compare results statistically ranks relative clone size between genotypes rather than using the absolute values of the estimates.

Many of the genetic backgrounds in which mosaics were generated carried other, distant *FRT* sites as part of the *FRT82B Xrp1$^{m2-73}$*, *FRT82B RpS3*, *rpS12$^{G97D}$ FRT80B, Df(3L)H99 FRT80B* chromosomes. Accordingly, the control backgrounds in these experiments always included *FRT82B* or *FRT80B*, as appropriate, and as described in the figure legends.

The *RpL28* rescue transgene was obtained by inserting genomic sequences 3L: 3220152–3225729 (*Drosophila* genome Release 6) into pTL780, which uses DsRed expression as a transgenic marker (**Blanco-Redondo and Langenhan, 2018**). The genomic DNA was amplified from the

*Drosophila* genomic reference strain (*Adams et al., 2000*). The resulting pTL780(RpL28+) plasmid was used for integration at the VK37 landing site on chromosome 2 (*Venken et al., 2009*).

For irradiation, food vials containing larvae were exposed to 500, 1000 or 4000 rad from a $\gamma$-ray source 84 ± 12 hr after egg laying. Dissection, fixation, and immuno-labeling of wing imaginal discs with anti-active Dcp1 and anti-Xrp1 was performed as described previously (*Baker et al., 2014*; *Lee et al., 2018*).

## Statistics

Frequencies of cell death and of Xrp1 expression were compared pairwise by t-tests (*Figure 1*). For multiple comparisons, one-way ANOVA was used with the Holm correction for multiple testing (*Figure 1*, *Figure 10*). Previous studies indicated that significant results could be obtained from five biological replicates, where a biological replicate is an imaginal disc preparation labeled, imaged, and quantified (*McNamee and Brodsky, 2009*). N for each experiment is reported in the figure legends. The extent of white tissue in mosaic eyes was compared using pairwise Mann-Whitney tests with the Benjamini-Hochberg (BH) correction for multiple testing, using FDR $\leq$ 0.05. There are 109 pairwise Mann-Whitney comparisons made in the main text of this paper, their P-values and the BH corrections are summarized in *Supplementary file 2*. Where the extent of white tissue in mosaic eyes was compared between multiple genotypes simultaneously, the Kruskal-Wallis test was used with post-hoc follow-up tests using the method of Conover with BH correction using FDR $\leq$ 0.05. No explicit power analysis was used. All flies obtained were scored in initial experiments, sometimes leading to unequal sample sizes, subsequently we considered 20 eyes of each sex generally sufficient for significant results (while the number of flies that can be obtained is rarely limiting, blinding and scoring clone sizes is time-consuming). N is given in the figures for each experiment. All the figures show experimental and control data obtained from simultaneous parallel experiments in each case, for which all the data scored were included. Some of the genotypes have been generated on multiple occasions with similar results, not all included in the figures.

## Acknowledgements

We thank Jorge Blanco, Michael Brodsky, Kevin Cook, Kent Golic, and Cristina Montagna for useful discussions, Tao Wang for statistical advice, and D Rio for Xrp1-specific antibodies. This study would not have been possible without genetic strains obtained from the Exelixis Collection at Harvard Medical School and from the Bloomington *Drosophila* Stock Center (supported by NIH P40OD018537). We also thank Erika Bach, Susan Celniker, and Gunter Reuter for genetic strains. We thank S Emmons, J Hebert, A Jenny, M Kiparaki, A Kumar, C Montagna, J Secombe, and A Tomlinson for comments on this or earlier versions of the manuscript. Supported by a grant from the NIH (GM104213). Confocal Imaging was performed at the Analytical Imaging Facility, Albert Einstein College of Medicine, supported by NCI cancer center support grant (P30CA013330), using Leica SP5 and SP8 microscopes, the latter acquired through NIH SIG 1S10 OD023591. This paper includes data from a thesis partially fulfilling of the requirements for the Degree of Doctor of Philosophy in the Graduate Division of Medical Sciences, Albert Einstein College of Medicine, Yeshiva University.

## Additional information

### Funding

| Funder | Grant reference number | Author |
| --- | --- | --- |
| National Institute of General Medical Sciences | GM104213 | Nicholas Baker |
| National Cancer Institute | P30CA013330 | Nicholas Baker |
| NIH | SIG 1S10 OD023591 | Nicholas Baker |

The funders had no role in study design, data collection and interpretation, or the decision to submit the work for publication.

## Author contributions
Zhejun Ji, Conceptualization, Data curation, Formal analysis, Investigation, Visualization, Methodology, Writing - original draft; Jacky Chuen, Data curation, Investigation; Marianthi Kiparaki, Conceptualization; Nicholas Baker, Conceptualization, Data curation, Formal analysis, Supervision, Funding acquisition, Validation, Investigation, Visualization, Methodology, Writing - original draft, Project administration, Writing - review and editing

## Author ORCIDs
Jacky Chuen (iD) https://orcid.org/0000-0003-4781-6907
Nicholas Baker (iD) https://orcid.org/0000-0002-4250-3488

## Decision letter and Author response
Decision letter https://doi.org/10.7554/eLife.61172.sa1
Author response https://doi.org/10.7554/eLife.61172.sa2

## Additional files

### Supplementary files

• Supplementary file 1. FRT insertions used in this study and their combinations. Columns indicate the specific insertion, genome location, and cytological position of the elements used as the left FRT site. Rows indicate the same information for the right FRT site. Insertions in the 'd' family are of the P{XP} element, the 'e' family the PBac{RB} element, and 'f' family PBac{WH} (*Thibault et al., 2004*). For clarity, in the main text we refer to genetic strains by the cytological insertion point eg '63A3-65A9' is the shorthand descriptive name for the PBac{WH}$^{f01922}$ P{XP}$^{d02570}$ chromosome. Which *Rp* genes are included between FRT sites is shown, as is *eIF2γ*. Some FRT combinations were poor substrates for Flp, either retained the parental eye color in the presence of eyFlp, or produce a salt and pepper pattern of very small clones that is indicative of excision occurring only late in development once large cell numbers are present (*Figure 2U–Z*). These results are summarized by shading FRT combinations tested as follows: Green – eyFlp recombination in essentially all cells; Blue – eyFlp recombination in most cells, associated with small eye size ($\leq 0.5$ linear dimensions); magenta – eyFlp recombination not detected; Orange – eyFlp recombination only late in development gives a mottled eye. The interpretation that recombination is reduced or absent is preferred to the alternative possibility that excision results in a cell-lethal genotype that later disappears, in part because results correlated with individual FRT elements and not with the genetic material between them. For example, recombination between 26A1 and 28F3 or 29C1, revealed only small, late recombination, but the 26A1-29F8 recombination that deletes all the same sequences was completely excised from EyFlp eyes and developed normal eye size with entirely Df(2L)26A1-29F8/+ cells (*Figure 2D*). Other examples of recombinations that could not readily be obtained were 87B8-89B18, 87B8-91B8, 87B8-92F1 and 89B13-92F1, although the larger 89B8-93A2 and 89B13-93A2 recombinations were readily obtained (*Figure 2R–T*). In contrast to the lack of correlation with deleted chromosome regions, when an element was not recombined by eyFLP this was the case with all the partner elements tested, so each FRT element could be designated as green or orange/magenta without ambiguity (for the 21-23/4 region elements there is insufficient information to identify the particular non-recombining elements). These data suggest that some *FRT*-containing Exelixis elements are poor substrates for cis-recombination in the head. Interestingly, all 7 insertions of the PBac (*Adams and Cory, 1998*) element tested belong in this category, although this element has previously been recombined successfully in the germline (*Parks et al., 2004*).

• Supplementary file 2. Statistical comparisons of segmental-aneuploid cell contribution to adult eyes. 109 pairwise comparison between segmental aneuploid genotypes were performed in this study, which requires multiple testing correction. The table shows each comparison ranked according to raw p-value (Mann-Whitney), Benjamini-Hochberg critical value for FDR $\leq$ 0.05, adjusted p-value, and significance.

• Transparent reporting form

Data availability

All data generated or analysed during this study are included in the manuscript and supporting files.

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

# Appendix 1

**Appendix 1—key resources table**

| Reagent type (species) or resource | Designation | Source or reference | Identifiers | Additional information |
|---|---|---|---|---|
| Gene (*Drosophila melanogaster*) | RpL28 | | Flybase: FBgn0035422 | |
| Gene (*Drosophila melanogaster*) | eIF2γ | | Flybase: FBgn0263740 | |
| Genetic reagent (*D. melanogaster*) | w[11-18] | *Hazelrigg et al., 1984*; *Lee et al., 2016* | FLYBASE: FBal0018186 | Bloomington *Drosophila* Stock Center #3605 |
| Genetic reagent (*D. melanogaster*) | Xrp1[m2-73] | *Lee et al., 2016* | FLYBASE: FBal0346068 | Bloomington *Drosophila* Stock Center #81270 |
| Genetic reagent (*D. melanogaster*) | rpS12[G97D] | *Tyler et al., 2007* | FLYBASE: FBal0193403 | |
| Genetic reagent (*D. melanogaster*) | (Df3L)H99 | *Abbott and Lengyel, 1991* | FLYBASE: FBab0022359 | Bloomington *Drosophila* Stock Center #1576 |
| Genetic reagent (*D. melanogaster*) | (M3R)w[124] aka RpS3[2] | *Ferrus, 1975*; *Abbott and Lengyel, 1991* | FLYBASE: FBal0011951 | |
| Genetic reagent (*D. melanogaster*) | hs-FLP | *Struhl and Basler, 1993* | FLYBASE: FBtp0001101 | |
| Genetic reagent (*D. melanogaster*) | ey-FLP | *Newsome et al., 2000* | FLYBASE: FBal0098303 | |
| Genetic reagent (*D. melanogaster*) | P{neoFRT}80B | *Xu and Rubin, 1993* | FLYBASE: FBti0002073 | Bloomington *Drosophila* Stock Center #1988 |
| Genetic reagent (*D. melanogaster*) | P{neoFRT}82B | *Xu and Rubin, 1993* | FLYBASE: FBti0002074 | Bloomington *Drosophila* Stock Center #2050, 2051 |
| Genetic reagent (*D. melanogaster*) | P{rpS12+8 kb} | *Xu and Rubin, 1993*; *Kale et al., 2018* | FLYBASE: FBal0337985 | |
| Genetic reagent (*D. melanogaster*) | P{rpS12-G97D8kb} | *Xu and Rubin, 1993*; *Kale et al., 2018* | FLYBASE: FBal0337986 | |
| Genetic reagent (*D. melanogaster*) | P{ry[+] Su(var)3–9[+] eIF2γ[+]} | *Xu and Rubin, 1993*; *Tschiersch et al., 1994*; *Kale et al., 2018* | | |
| Genetic reagent (*D. melanogaster*) | P{DsRed; RpL28[+]} | This study | | See Materials and methods; Dr. Nicholas Baker's lab. |
| Genetic reagent (*D. melanogaster*) | P53[5A-1-4] | *Xie and Golic, 2004* | FLYBASE: FBal0138188 | Bloomington *Drosophila* Stock Center #6815 |

*Continued on next page*

*Appendix 1—key resources table continued*

| Reagent type (species) or resource | Designation | Source or reference | Identifiers | Additional information |
| --- | --- | --- | --- | --- |
| Genetic reagent (*D. melanogaster*) | PBac{RB}CG11617$^{e00462}$ | *Xu and Rubin, 1993; Thibault et al., 2004* | FLYBASE: FBal0162546 | Bloomington *Drosophila* Stock Center #17859 |
| Genetic reagent (*D. melanogaster*) | PBac{WH}MED15$^{f04180}$ | *Xu and Rubin, 1993; Thibault et al., 2004* | FLYBASE: FBti0042319 | Bloomington *Drosophila* Stock Center #18739 |
| Genetic reagent (*D. melanogaster*) | P{XP}CG9016$^{d08241}$ | *Xu and Rubin, 1993; Thibault et al., 2004* | FLYBASE: FBal0160858 | Bloomington *Drosophila* Stock Center #19290 |
| Genetic reagent (*D. melanogaster*) | P{XP}CG9003$^{d09761}$ | *Xu and Rubin, 1993; Thibault et al., 2004* | FLYBASE: FBal0160860 | Bloomington *Drosophila* Stock Center #19321 |
| Genetic reagent (*D. melanogaster*) | P{XP}salto$^{d09417}$ | *Xu and Rubin, 1993; Thibault et al., 2004* | FLYBASE: FBal0159854 | Bloomington *Drosophila* Stock Center #19315 |
| Genetic reagent (*D. melanogaster*) | P{XP}CG11200$^{d02302}$ | *Xu and Rubin, 1993; Thibault et al., 2004* | FLYBASE: FBal0162606 | Bloomington *Drosophila* Stock Center #19173 |
| Genetic reagent (*D. melanogaster*) | P{XP}sob$^{d06074}$ | *Xu and Rubin, 1993; Thibault et al., 2004* | FLYBASE: FBal0158622 | Bloomington *Drosophila* Stock Center #19230 |
| Genetic reagent (*D. melanogaster*) | PBac{WH}Uro$^{f04888}$ | *Xu and Rubin, 1993; Thibault et al., 2004* | FLYBASE: FBal0159557 | Bloomington *Drosophila* Stock Center #18814 |
| Genetic reagent (*D. melanogaster*) | PBac{RB}Ssb-c31a$^{e02272}$ | *Xu and Rubin, 1993; Thibault et al., 2004* | FLYBASE: FBal0159728 | Bloomington *Drosophila* Stock Center #18032 |
| Genetic reagent (*D. melanogaster*) | PBac{RB}CG31898$^{e03937}$ | *Xu and Rubin, 1993; Thibault et al., 2004* | FLYBASE: FBal0161732 | Bloomington *Drosophila* Stock Center #18211 |
| Genetic reagent (*D. melanogaster*) | PBac{WH}CG9582$^{f00857}$ | *Xu and Rubin, 1993; Thibault et al., 2004* | FLYBASE: FBal0160790 | Bloomington *Drosophila* Stock Center #18378 |
| Genetic reagent (*D. melanogaster*) | PBac{WH}tei$^{f00157}$ | *Xu and Rubin, 1993; Thibault et al., 2004* | FLYBASE: FBal0159247 | Bloomington *Drosophila* Stock Center #18299 |
| Genetic reagent (*D. melanogaster*) | PBac{RB}CG13018$^{e00535}$ | *Xu and Rubin, 1993; Thibault et al., 2004* | FLYBASE: FBal0162408 | Bloomington *Drosophila* Stock Center #17863 |
| Genetic reagent (*D. melanogaster*) | PBac{RB}Cpr51A$^{e03998}$ | *Xu and Rubin, 1993; Thibault et al., 2004* | FLYBASE: FBal0162723 | Bloomington *Drosophila* Stock Center #18221 |

*Appendix 1—key resources table continued*

| Reagent type (species) or resource | Designation | Source or reference | Identifiers | Additional information |
|---|---|---|---|---|
| Genetic reagent (*D. melanogaster*) | PBac{WH}CG10384[f04349] | *Xu and Rubin, 1993*; *Thibault et al., 2004* | FLYBASE: FBal0162694 | Bloomington *Drosophila* Stock Center #18762 |
| Genetic reagent (*D. melanogaster*) | PBac{WH}CG42260[f00464] | *Xu and Rubin, 1993*; *Thibault et al., 2004* | FLYBASE: FBal0225307 | |
| Genetic reagent (*D. melanogaster*) | PBac{WH}Jafrac2[f01922] | *Xu and Rubin, 1993*; *Thibault et al., 2004* | FLYBASE: FBal0160273 | Bloomington *Drosophila* Stock Center #18489 |
| Genetic reagent (*D. melanogaster*) | PBac{WH}CG17746[f05041] | *Xu and Rubin, 1993*; *Thibault et al., 2004* | FLYBASE: FBal0162020 | Bloomington *Drosophila* Stock Center #18834 |
| Genetic reagent (*D. melanogaster*) | P{XP}Leash[d06455] | *Xu and Rubin, 1993*; *Thibault et al., 2004* | FLYBASE: FBal0161383 | Bloomington *Drosophila* Stock Center #19240 |
| Genetic reagent (*D. melanogaster*) | P{XP}cu[d05983] | *Xu and Rubin, 1993*; *Thibault et al., 2004* | FLYBASE: FBal0158886 | Bloomington *Drosophila* Stock Center #19225 |
| Genetic reagent (*D. melanogaster*) | $w^{1118}$; P{XP}d06796/TM6B, $Tb^1$ | *Xu and Rubin, 1993*; *Thibault et al., 2004* | FLYBASE: FBti0042862 | Bloomington *Drosophila* Stock Center #19250 |
| Genetic reagent (*D. melanogaster*) | P{XP}CG10311[d06928] | *Xu and Rubin, 1993*; *Thibault et al., 2004* | FLYBASE: FBal0162706 | Bloomington *Drosophila* Stock Center #19255 |
| Genetic reagent (*D. melanogaster*) | P{XP}d02570 | *Xu and Rubin, 1993*; *Thibault et al., 2004* | FLYBASE: FBti0054904 | |
| Genetic reagent (*D. melanogaster*) | P{XP}wrm1[d02813] | *Xu and Rubin, 1993*; *Thibault et al., 2004* | FLYBASE: FBal0160902 | Bloomington *Drosophila* Stock Center #19182 |
| Genetic reagent (*D. melanogaster*) | P{XP}UGP[d07256] | *Xu and Rubin, 1993*; *Thibault et al., 2004* | FLYBASE: FBal0159573 | Bloomington *Drosophila* Stock Center #19267 |
| Genetic reagent (*D. melanogaster*) | PBac{WH}CG14894[f04937] | *Xu and Rubin, 1993*; *Thibault et al., 2004* | FLYBASE: FBal0162215 | Bloomington *Drosophila* Stock Center #18821 |
| Genetic reagent (*D. melanogaster*) | PBac{RB}Cad89D[e03186] | *Xu and Rubin, 1993*; *Thibault et al., 2004* | FLYBASE: FBal0160726 | Bloomington *Drosophila* Stock Center #18129 |
| Genetic reagent (*D. melanogaster*) | PBac{WH}Actn3[f00971] | *Xu and Rubin, 1993*; *Thibault et al., 2004* | FLYBASE: FBal0162831 | Bloomington *Drosophila* Stock Center #18397 |

*Appendix 1—key resources table continued*

| Reagent type (species) or resource | Designation | Source or reference | Identifiers | Additional information |
|---|---|---|---|---|
| Genetic reagent (*D. melanogaster*) | PBac{RB}qin$^{e03728}$ | *Xu and Rubin, 1993*; *Thibault et al., 2004* | FLYBASE: FBal0162275 | Bloomington *Drosophila* Stock Center #18186 |
| Genetic reagent (*D. melanogaster*) | PBac{RB} DPCoAC$^{e03144}$ | *Xu and Rubin, 1993*; *Thibault et al., 2004* | FLYBASE: FBal0175762 | Bloomington *Drosophila* Stock Center #18121 |
| Genetic reagent (*D. melanogaster*) | PBac{WH}KaiR1D$^{f03502}$ | *Xu and Rubin, 1993*; *Thibault et al., 2004* | FLYBASE: FBal0161451 | Bloomington *Drosophila* Stock Center #18663 |
| Genetic reagent (*D. melanogaster*) | PBac{WH}TotC$^{f01700}$ | *Xu and Rubin, 1993*; *Thibault et al., 2004* | FLYBASE: FBal0159656 | Bloomington *Drosophila* Stock Center #18460 |
| Antibody | polyclonalRabbit anti-XRP1(short) | *Francis et al., 2016* | | (1:200) dilution |
| Antibody | Polyclonal Rabbit anti-active-Dcp1 | Cell Signalling Technology | Cat #9578 | (1:50) dilution |
| Antibody | Polyclonal Donkey anti-Rabbit IgG, Cy3 conjugate | Jackson Immunoresearch | Cat # 711-165-152 | (1:200) dilution |
| Recombinant DNA reagent | P{DsRed; RpL28$^{+}$} | This study | | See Materials and methods; Dr. Nicholas Baker's lab. |
| Recombinant DNA reagent | pGE-attBTT-loxP-DsRed (pTL780) | *Blanco-Redondo and Langenhan, 2018* | | Addgene Plasmid #115160 |

