## [Decision Letter]

**Acceptance summary:**

Aneuploid cells are found in developing mammalian embryos but are not present in the adult. These results suggest that aneuploid cells are recognized and eliminated, but there have been no molecular insights into how such surveillance occurs. Dr. Baker and colleagues provide rigorous experiments in developing *Drosophila* tissues to show that cell competition eliminates aneuploid cells based on ribosomal protein (Rp) gene dosage. Since Rp genes are found throughout the genome, their results suggest that Rp genes are sentinels for genomic aberrations like aneuploidy. These results will likely have broad implications for vertebrate development and human tumors, which are frequently comprised of aneuploid cells.

**Decision letter after peer review:**

Thank you for submitting your article "Cell competition removes aneuploid cells from *Drosophila* imaginal discs based on ribosomal protein gene dose" for consideration by *eLife*. Your article has been reviewed by four peer reviewers, including Erika A Bach as the Reviewing Editor and Reviewer #1, and the evaluation has been overseen by Maureen Murphy as the Senior Editor. The following individual involved in review of your submission has agreed to reveal their identity: Laura Johnston (Reviewer #3).

The reviewers have discussed the reviews with one another and the Reviewing Editor has drafted this decision to help you prepare a revised submission.

Summary:

Li and colleagues test the role of cell competition in removing segmental aneuploid cells. They make chromosomal deletions in the developing *Drosophila* eye imaginal disc that either include or exclude a ribosomal (Rp) locus. They find that segmental aneuploid cells heterozygous for an Rp gene are eliminated while those that are +/+ for Rp genes are not. They show that the elimination of Rp/+ segmental aneuploid cells requires ribosomal protein S12 and the transcription factor Xrp1, which the Baker lab previously found was required to remove suboptimal (Rp/+) cells from developing tissues through cell competition. These results lead the authors to conclude that cell competition performs a surveillance function to remove potentially dangerous aneuploid cells based on the cell's Rp gene dosage. They try to extend these results to the late, p53-independent cell death after ionizing radiation (IR), but more data are needed to prove this. The strengths of the study include elegant and classical genetic approaches using FLP/FRT to make defined chromosomal deletions and an impressive number of genetic backgrounds in which to test them. The weaknesses include lack of proof that cell competition causes the late cell death after IR, testing a comparatively small number of aneuploidies that do not remove Rp genes which potentially skews the results, lack of confidence in stated methodology, overstating the results, and the need for extensive editorial changes.

Essential revisions:

1) The reviewers judged that the authors do not yet prove that the late cell death after IR occurs through cell competition because (a) the authors do not show that cells die because of their proximity to wild type cells and (b) removing both p53 and S12 does not block completely late cell death. To strengthen this conclusion, the authors could irradiate M/+ animals and look at death 4 hours post IR. If interpretable, this result may (1) solidify the conclusion that the late death is due to cell competition and (2) give more weight to their hypothesis that the increased cancer incidence in Diamond Blackfan Anemia (DBA) patients is because M/+ cells are not able to perform a surveillance function for aneuploid cells. The authors should perform this experiment (or one very similar to it) and temper their conclusions.

2) The number of aneuploidies that do not remove Rp genes was quite small (only 4 out of 17). Additionally, the deletions that remove Rp genes are always larger than those than do not remove Rp genes, raising alternative (i.e., non-cell competition) explanations about their elimination. In order to strengthen their major conclusion that aneuploid cells heterozygous for an Rp gene are eliminated by cell competition, the authors ideally should test additional deletions that do not remove an Rp gene. Since this may not be feasible, the authors should at least moderate their conclusions.

3) Methodology:

a) The authors need to address how they performed the percentage white assay. Specifically, they need explain whether the area occupied by white (unpigmented) ommatidia was assigned by observation on the part of the experimentalist using a dissecting scope or was made by measurements using micrographs and software like ImageJ. This is a key assay in the paper and it is not clear how an experimentalist using a dissecting scope could determine whether white ommatidia represent a precise value (e.g. 46%) of the eye tissue.

b) The authors need to better describe how they induced hs-FLP clones (stage of development when clone was induced, etc).

4) The manuscript requires extensive editorial changes. These include:

a) Abstract: should be re-phrased to remove overstatements that cell competition is the major driver of aneuploidy-induced cell death.

b) Title: should be rephrased so that it does not imply that all aneuploid cells are removed as a consequence of ribosomal protein gene dose. This is a suggested title from the reviewers ""Cell competition in *Drosophila* imaginal discs removes cells with small aneuploidies based on ribosomal protein gene dose"

c) Introduction:

– include publications demonstrating that the death of aneuploid cells in the wing disc is p53-independent (for example, Dekanty et al., 2012 and Morais da Silva et al., 2013) and provide a more thorough explanation of IR-induced aneuploidies, citing the original Brodsky paper and more recent publications based on it.

– reference the literature of aneuploidy-induced stresses (see review Zhu et al., 2018).

– modify the last paragraph to reflect the fact that removing both p53 and S12 does not block completely late cell death.

d) Results:

– revise the paragraph describing the "y" bristle

– revise the paragraph about Xrp1 and about RpS3

– state that one cannot conclude whether aneuploid cells not lacking an Rp gene grow at the same rate as wild type cells

– clarify how much of the genome was tested in each aneuploid condition as suggested by the reviewers, who indicated that 2-4% of the genome was tested, and make clear that segmental aneuploidies are rather small.

e) Discussion:

– remove redundancies and reduce the semblance of a review article.

– mention that segmental aneuploidies generated in this study are comparatively smaller in size than aneuplodies generated in other model organisms and/or in other *Drosophila* studies.

– when talking about the role of Xrp1 and S12 in IR-induced late cell death, state that these results "suggest" that cell competition is in action.

– edit the Discussion to reflect the fact that DBA patients have higher rate of hematopoietic malignancies. Their proposed aneuploidy surveillance mechanism works well for epithelial tumors, but it is not clear how this would function in blood cancers.

5) Figures:

a) The authors should consider which figures could be moved to supplementary (e.g., the y bristle figure)

b) Figure 1: The authors need to explain why did not perform experiments with S12 mutation alone. If possible, they should add these data.

6) Tables: All of the raw data should be put into a new supplementary file.

7) Statistics: the authors need to address how their statistical analyses accounted for the large variation in sample sizes.

---

## [Author Response]

Essential revisions:1) The reviewers judged that the authors do not yet prove that the late cell death after IR occurs through cell competition because (a) the authors do not show that cells die because of their proximity to wild type cells and (b) removing both p53 and S12 does not block completely late cell death. To strengthen this conclusion, the authors could irradiate M/+ animals and look at death 4 hours post IR. If interpretable, this result may (1) solidify the conclusion that the late death is due to cell competition and (2) give more weight to their hypothesis that the increased cancer incidence in Diamond Blackfan Anemia (DBA) patients is because M/+ cells are not able to perform a surveillance function for aneuploid cells. The authors should perform this experiment (or one very similar to it) and temper their conclusions.

a) We agree that we did not formally prove that late cell death after IR occurs via cell competition, since we did not demonstrate a requirement for proximity to wild type cells. The reviewers suggested measuring late cell death in the background of a germline *Rp* mutant, since this could demonstrate a requirement for *Rp^+/+^* cells in the mechanism. We had avoided this experiment previously because: 1) *Rp* mutant strains exhibit an elevated background of cell death that is not related to cell competition that might interfere; 2) *Rp* mutant strains exhibit a developmental delay. Although it is possible to adjust the time of irradiation so that it occurs at the same developmental stage, it is not then possible to measure cell death at the same time post-irradiation and know that this corresponds to the same developmental timepoint.

Given the enthusiasm of the reviewers, we constructed an *RpS18p53* double mutant strain, with some difficulty, and performed the requested experiment. The results were shared with the editors.

The expected result, if p53-independent cell death is cell competition requiring proximity to *Rp^+/+^* cells, is that less cell death should be seen in irradiated *RpS18^+/-^p53^-/-^* wing discs compared to *p53^-/-^*. If the hypothesis is false, the expected result is that levels of cell death should be the same. Increased cell death in irradiated *RpS18^+/-^p53^-/-^* is unexpected in either model and can’t yet be interpreted with respect to the cell competition hypothesis. It suggests there may be a distinct susceptibility to radiation in the *RpS18^+/-^* genotype, related to another process occurring in this genotype. Many more experiments would be necessary, however, to draw this conclusion. At a minimum, we would need to perform time-courses of the p53-dependent and p53independent cell death in the *RpS18^+/-^p53^-/-^*, to evaluate the cumulative amount of cell death and to assess whether, for example, the developmental delay difference means that the 24h post-irradiation timepoints are not comparable. Since we have already taken more than twice the normal time allotted for revisions to analyze this unhealthy genotype, however, and since the results make it unclear that this experiment can address the original hypothesis, we suggest that this line of experiments is not worth continuing at present. As this experiment was uninformative, we have revised the manuscript to make it clear that it cannot be considered proven that cell competition is responsible for the delayed, p53-independent cell death. Our results still represent significant progress, because we have shown that much of this cell death shares genetic requirements with cell competition, which is consistent with cell competition being responsible. We make it clear in the revised manuscript that the section on the p53-independent cell death served as a motivation for the subsequent studies of precisely-defined genetic damage, which are more definitive.

b) We do not understand why the reviewers think it so significant that late cell death is blocked incompletely in *p53 rps12* double mutants. Our results show that the majority of the late cell death has properties consistent with cell competition. Our conclusions are not affected if there is also another component of late cell death. We now temper our conclusions, however, by mentioning that the RpS12-independent process might not be cell competition. The explanation could be as simple as that the *rpS12* mutant allele allows a small amount of cell competition to continue.

2) The number of aneuploidies that do not remove Rp genes was quite small (only 4 out of 17). Additionally, the deletions that remove Rp genes are always larger than those than do not remove Rp genes, raising alternative (i.e., non-cell competition) explanations about their elimination. In order to strengthen their major conclusion that aneuploid cells heterozygous for an Rp gene are eliminated by cell competition, the authors ideally should test additional deletions that do not remove an Rp gene. Since this may not be feasible, the authors should at least moderate their conclusions.

The number of aneuploidies that do not remove *Rp* genes is 5 (actually 6, but two are overlapping). We now mention the reviewers’ suggestion that there could be something exceptional about these regions in the revised manuscript, but only to argue against this idea. If there are regions of the genome with unusual properties, then it is not likely that we would have picked these regions. If the other, non-Rp mechanisms were widespread, then they would be likely have been shown by some of the regions we examined. The likelihood that there is a widespread mechanism for removing monosomic genotypes that has not been encountered in any of 5 randomly-selected genomic regions is therefore not very high. It is worth mentioning that we have tried to select the largest non-Rp regions possible for analysis. This is for the practical reason that we need to select germline recombinants between two linked transgenes in setting up these experiments, and this is easier to do when they map further apart. This is probably how we managed to find eIF2g, one of the few non-Rp genes causing cell competition, because we picked a large region lacking any Rp gene for study, which in fact contained the eIF2g locus. Our data suggest that if there are such *Rp* gene-free regions with different properties, they are in the minority.

The reviewers’ suggest that the *Rp*-containing deletions are in fact eliminated because they are larger and not because they contain *Rp* genes. This is not supported by the existing data. It is true that Df(3L)63A3-65A9, which is eliminated, is larger than Df(3L)63C1-65A9, which is not, but we show clearly by germline transformation that this difference is mediated by a single locus, *RpL28*. In ongoing experiments in the lab, we have shown that a second region also is eliminated because of a single Rp gene, not because it is larger. We also show that Df(3R)87B8-89B16 and Df(3R)87B8-89B5 cells are eliminated because of a single gene, eIF2g. Thus, in 3 out of 3 cases, we demonstrate directly that single genes led to elimination of particular monosomies are eliminated, not size.

We agree that it is possible that there is a cumulative effect of gene dose that only becomes apparent at still larger monosomy sizes and can also cause cell competition independently of *Rp* loci. Accordingly, we included this possibility in the revised manuscript. If correct, larger monosomies would then be eliminated both because of *Rp* loci and the cumulative effect of other loci, ie it is not correct to state they would not be subject to *Rp*-dependent cell competition.

We do not understand how non-cell competition mechanisms could be proposed for removal of any of these aneuploidies. Figure 2 showed that all 17 genotypes survive when the whole eye is mutant, thus competition with wild type cells is required for their loss. It seems possible there is a typo in the review summary and that “non-Rp cell competition” was the intended meaning. If so, this would be addressed by the preceding paragraph.

3) Methodology:a) The authors need to address how they performed the percentage white assay. Specifically, they need explain whether the area occupied by white (unpigmented) ommatidia was assigned by observation on the part of the experimentalist using a dissecting scope or was made by measurements using micrographs and software like ImageJ. This is a key assay in the paper and it is not clear how an experimentalist using a dissecting scope could determine whether white ommatidia represent a precise value (e.g. 46%) of the eye tissue.

The revised manuscript explains more clearly that this was done manually by an experimenter. We agree that the measurements (like any measurements) will be subject to a margin of error. We do not think that the error needs to be large or significant. Does the reviewer not agree that a reasonably careful observer could distinguish between an eye that was 1/5 white and an eye that was 1/6 white? That is a 3% difference (20% vs 17%). We have done the controls of scoring samples by different investigators, or scoring the same samples on different occasions, and do not think that large errors are involved. It is worth mentioning (and included in the revised manuscript) that the Mann Whitney procedure used to evaluate results statistically compares relative clone size, not absolute measurements. Finally, if our measurements were error-prone, the result would be that statistical differences between samples would be harder to obtain, because they would be randomized by the errors. Random errors cannot easily lead to significant differences between genotypes when the genotypes are scored blind, unlike what we find.

In our opinion, this is a robust assay, the most quantitative, convenient and reliable assay yet devised for cell competition, and likely to be adopted rapidly throughout the field once it is available.

It may be useful to develop a digital scoring assay in future, although there will be challenges. Since the fly eye is not flat, eye clone photography is challenging because of depth of field (not all the eye is in focus), also parts of the eye are always tilted away from the camera, distorting clone area. Addressing these issues might require software development. Preparing the eye clone photographs for this paper using conventional methods was in fact very time-consuming.

b) The authors need to better describe how they induced hs-FLP clones (stage of development when clone was induced, etc).

How the white clones were induced by hsFlp is described in the Materials and methods.

4) The manuscript requires extensive editorial changes. These include:a) Abstract: should be re-phrased to remove overstatements that cell competition is the major driver of aneuploidy-induced cell death.

The revised Abstract has been modified to state only that removal of damaged cells often required cell competition genes, rather than concluding that cell competition itself has been demonstrated. Because of the strict 150 word limit, adding this caveat required removing other information from the Abstract, such as that irradiation and FLP-FRT recombination were used, and that aneuploidy is deleterious in humans.

We are not sure this was the right choice.

b) Title: should be rephrased so that it does not imply that all aneuploid cells are removed as a consequence of ribosomal protein gene dose. This is a suggested title from the reviewers ""Cell competition in *Drosophila* imaginal discs removes cells with small aneuploidies based on ribosomal protein gene dose"

We revised the title to refer to segmental aneuploidies. We would like to also replace “removes” with “can remove”, but this is not possible because it exceeds the character limit (by 1 character). We prefer not to use the word “small”, which is subjective (see discussion under part d below).

c) Introduction:– include publications demonstrating that the death of aneuploid cells in the wing disc is p53-independent (for example, Dekanty et al., 2012 and Morais da Silva et al., 2013) and provide a more thorough explanation of IR-induced aneuploidies, citing the original Brodsky paper and more recent publications based on it.– reference the literature of aneuploidy-induced stresses (see review Zhu et al., 2018).– modify the last paragraph to reflect the fact that removing both p53 and S12 does not block completely late cell death.

Additional publications demonstrating the p53-independent death of aneuploid cells are included in the revised Introduction. More details of IR-induced aneuploidies are now given, including the Brodsky 2004 and 2009 papers. The introduction to aneuploidy-induced stresses has been expanded, also the Discussion. We modified the last Introduction paragraph so that it states only that most late IR-induced cell death resembles cell competition genetically.

d) Results:– revise the paragraph describing the "y" bristle– revise the paragraph about Xrp1 and about RpS3– state that one cannot conclude whether aneuploid cells not lacking an Rp gene grow at the same rate as wild type cells– clarify how much of the genome was tested in each aneuploid condition as suggested by the reviewers, who indicated that 2-4% of the genome was tested, and make clear that segmental aneuploidies are rather small.

We made minor revisions to the paragraph describing IR-induced y bristles. We have shortened and clarified the Xrp1 paragraph. The possibility that aneuploid cells grow less well than wild type cells is mentioned in the revised Discussion. We discuss the size of our segmental aneuploid regions and compare them to aneuploidies in other studies. We prefer not to say that they are “rather small”. We explain that most of the aneuploidies studied are as large or larger in genetic content than some whole human chromosomes. The complete genomic information is given in Supplementary file 1. We state how much of the genome was tested. The salient fact here is that 80% of the *Drosophila* genome is carried on 2 pairs of autosomes, so that whole chromosome aneuploidies in *Drosophila* mimic very abnormal human karyotypes affecting ~10 chromosomes at once. We prefer to give these specifics rather than to use the adjective “small”, which could be interpreted as “unimportant”. We note that the other papers that the reviewers’ want cited have not clarified that they only studied “large” aneuploidies.

e) Discussion:– remove redundancies and reduce the semblance of a review article.– mention that segmental aneuploidies generated in this study are comparatively smaller in size than aneuplodies generated in other model organisms and/or in other *Drosophila* studies.– when talking about the role of Xrp1 and S12 in IR-induced late cell death, state that these results "suggest" that cell competition is in action.– edit the Discussion to reflect the fact that DBA patients have higher rate of hematopoietic malignancies. Their proposed aneuploidy surveillance mechanism works well for epithelial tumors, but it is not clear how this would function in blood cancers.

We have removed some redundancy from the Discussion, but still think it important that the Discussion include some summary of the Results. Since the Results section contains rather technical *Drosophila* genetics, we imagine that readers from other fields may skip to the Discussion, and we wish to keep the article accessible for such readers. We hope the editors will agree. We are not sure what “resemblance to a review article” means. The reviewers have in fact requested that we expand the discussion of other studies, some of which we do not think strictly relevant, for example studies of aneuploidy in *Drosophila* cells with chromosome instability, where the actual genotypes and role of competition are unknown. We have completely omitted many interesting topics, such as potential implications of our study for p53 function in cell competition, and in fact do intend to publish a follow-up review article addressing such topics. We acknowledge that the final paragraph is speculative, but this paragraph makes an important prediction about the potential role of cell competition in human cancer surveillance. We are surprised if the Editors would not like this prediction associated with the e*Life* paper. After all, it may prove to be correct. We have moderated the discussion of roles of Xrp1 and RpS12 in post-irradiation cell death to state that their requirements resemble those of cell competition. We do not understand the reviewers’ comments about tumors in DBA. In fact there is no enrichment for hematological malignancy in DBA, the cancer predisposition appears to affect all sites (eg PMID: 30266775; 22362038). We are not certain that the tissue architecture of Hematopoietic Stem Cells in situ is well enough known to say how epithelial it is. In any case, cell competition is well established in the hematopoietic system, contradicting the reviewers’ assumption (eg PMID: 20208998; 20362536; 24828041).

5) Figures:a) The authors should consider which figures could be moved to supplementary (e.g., the y bristle figure)

The y bristles figure represents only a single panel (Figure 1S), 1/19 of a figure. We have no replacement to occupy the space if it this panel is removed, so no advantage accrues to making Figure 1S supplemental.

b) Figure 1: The authors need to explain why did not perform experiments with S12 mutation alone. If possible, they should add these data.

Experiments with the S12 mutation alone are in fact shown for multiple experiments in this figure (panels 1B, 1F, 1J). Perhaps the review means that we should have quantified cell death in RpS12 alone? Presumably the intention is to address whether the p53-independent. RpS12-dependent cell death also occurs in the presence of wild type p53? This is indeed an interesting question, but unfortunately in the presence of wild type p53 the amount of cell death observed is simply too high to be quantified. Even if we did find a way to quantify the massive number of dead cells clumped together in these discs, we have no confidence at all they are not affected by, for example, saturation of the corpse disposal mechanisms. We discussed this in the figure legend. The radiation dose chosen to maximize the detection of the p53-dependent cell death (4000 rad) is very high. If we study lower IR doses the number of p53-independent deaths may become quite low. We are currently preparing another paper for publication that directly addresses the potential roles of RpS12 and Xrp1 in the DNA damage response (rather than cell competition) and it seems better to address this question there.

6) Tables: All of the raw data should be put into a new supplementary file.

We added nine source files containing the raw data for all the quantitative assays.

7) Statistics: the authors need to address how their statistical analyses accounted for the large variation in sample sizes.

The Mann Whitney procedure can lose sensitivity if sample sizes are unequal. This would result in a failure to detect significance. We mention sample sizes in the revised manuscript.